# ERK1/2 is an ancestral organising signal in spiral cleavage

Océane Seudre [1,2], Allan M. Carrillo-Baltodano [1,2], Yan Liang[1] & José M. Martín-Durán [1✉]

Animal development is classified as conditional or autonomous based on whether cell fates are specified through inductive signals or maternal determinants, respectively. Yet how these two major developmental modes evolved remains unclear. During spiral cleavage—a stereotypic embryogenesis ancestral to 15 invertebrate groups, including molluscs and annelids—most lineages specify cell fates conditionally, while some define the primary axial fates autonomously. To identify the mechanisms driving this change, we study *Owenia fusiformis*, an early-branching, conditional annelid. In *Owenia*, ERK1/2-mediated FGF receptor signalling specifies the endomesodermal progenitor. This cell likely acts as an organiser, inducing mesodermal and posterodorsal fates in neighbouring cells and repressing anteriorising signals. The organising role of ERK1/2 in *Owenia* is shared with molluscs, but not with autonomous annelids. Together, these findings suggest that conditional specification of an ERK1/2+ embryonic organiser is ancestral in spiral cleavage and was repeatedly lost in annelid lineages with autonomous development.

[1] School of Biological and Behavioural Sciences. Queen Mary University of London, Mile End Road, E1 4NS London, UK. [2]These authors contributed equally: Océane Seudre, Allan M. Carrillo-Baltodano. ✉email: chema.martin@qmul.ac.uk

The commitment of the first embryonic cells to more restricted developmental fates (e.g., endoderm, neuroectoderm and mesoderm) is a pivotal step in animal embryogenesis that leads to the establishment of body plans and influences subsequent development[1–3]. To define this early spatial organisation, animal embryos combine conditional, inductive cell–cell interactions with the asymmetric inheritance of cell-autonomous maternal determinants[1,2,4]. Frequently, one of these developmental strategies is predominant and thus animal embryogenesis is defined as either conditional or autonomous[1,2,4]. During evolution, animal lineages have transitioned between these two major modes of cell fate specification[5]. However, how these developmental transitions occur is unclear because they often coincide with additional variation in early embryogenesis (e.g., in cleavage patterns[6,7]) that makes the causes of cell fate specification shifts difficult to identify.

Spiral cleavage is an ancient and stereotypic early developmental programme characterised by alternating oblique cell divisions from the 4-cell stage onwards that is found among invertebrate groups within Spiralia, including molluscs and annelids[8,9] (Fig. 1a). Spiral cleaving embryos organise cell fates around four embryonic quadrants—named A–D—that roughly correspond to the left, ventral, right and dorsal body sides, respectively[8,9]. Although spiral cleavage is often depicted as a textbook example of autonomous development[2,4], spiral cleaving embryos specify their axial identities either conditionally or autonomously, without this affecting the overall conservation of the cleavage programme and the embryonic cell lineages[8–10] (Fig. 1b). In the conditional mode of cell fate specification, inductive signals between the cells in the animal pole and one vegetal blastomere at ~32–64-cell stage commit the latter to the dorsal D fate[11]. This cell will act as embryonic organiser, instructing neighbouring cells towards certain fates and establishing the animal body plan (Fig. 1b). This mode of spiral cleavage is widespread and most likely ancestral to Spiralia[11–13] (but see Dohle for an alternative hypothesis[14]) (Fig. 1c). However, some molluscan and annelid lineages[11,12] specify the axial identities through the asymmetric segregation of maternal determinants to one blastomere in the 4-cell stage embryo[15–17] (Fig. 1b). This blastomere will adopt the dorsal D fate, and one of its descendants will subsequently act as embryonic organiser. Therefore, the presence of both conditional and autonomous modes of development in spiral cleaving animals[12] is an ideal system to identify the cellular and molecular mechanisms underpinning cell fate specification transitions in animal embryos. Yet the mechanisms regulating spiral cleavage, and by extension the switches in developmental mode, are poorly understood.

The ERK1/2 signalling pathway—an evolutionarily conserved intracellular cascade of kinases[18]—specifies the dorsal D-quadrant identity and controls organising activity in Mollusca (Fig. 1c; Supplementary Table 1). In many conditional molluscs, inductive interactions activate ERK1/2 in the blastomere that adopts the dorsal D fate and acts as embryonic organiser[19–23], albeit the lineage identity of that cell varies across species (Supplementary Table 1). Likewise, active di-phosphorylated ERK1/2 is required in the D-quadrant blastomere functioning as embryonic organiser[24] in the otherwise autonomous mollusc *Tritia obsoleta*[17]. In Annelida however, autonomous species exhibit diverse patterns of ERK1/2 activity and do not require this signalling pathway to specify the dorsal D-quadrant and the embryonic organiser[25–28] (Fig. 1c; Supplementary Table 1). Yet the role of ERK1/2 during spiral cleavage is unknown for conditional annelids, as well as for most other spiralian groups (Fig. 1c). This prevents inferring the ancestral mechanisms controlling body patterning in Annelida, and Spiralia generally

(Fig. 1c), and thus it is unknown whether the axial organising role of ERK1/2 is a molluscan innovation or an ancestral cell fate specification mechanism lost in some autonomous annelid lineages.

In this work, we strategically studied the species *Owenia fusiformis*[29,30] to determine the role of ERK1/2 signalling in a conditional annelid (Fig. 1c). This marine species belongs to Oweniida—the sister group to all remaining annelids[31,32]—and exhibits embryonic traits considered ancestral to Annelida[30,33]. Therefore, the study of *O. fusiformis* can, by comparison with other annelid and spiralian lineages, help to infer ancestral traits to this group and Spiralia generally.

## Results

**Di-phosphorylated ERK1/2 is enriched in the 4d cell**. Unlike species with autonomous mode of development in which the blastomere of the D lineage that inherits maternal determinants is larger, the four embryonic quadrants in embryos with the conditional mode of spiral cleavage are symmetrical, which hinders inferring cell identities during early development. The cell acting as embryonic organiser in the D-quadrant, however, defers cell cycle progression compared to equivalent blastomeres in the A–C quadrants and exhibits enrichment of active (i.e., di-phosphorylated) ERK1/2 in some conditional molluscs[10,20] and putatively the conditional annelid *Hydroides hexagonus*[20] (Fig. 1c). Therefore, to determine the D lineage and identify a potential embryonic organiser in *O. fusiformis*, we first characterised the cell division dynamics and patterns of ERK1/2 activity in the vegetal pole from fertilisation to the onset of gastrulation (6 h post fertilisation, hpf). In this annelid, the first cleavage asymmetry occurs with the appearance of the fourth micromere quartet (4q cells) at ~4 hpf, when a pair of 4q cells forms before the other two (Supplementary Fig. 1a, b). Yet the vegetal pole remains symmetrical and embryonic quadrant identities are indiscernible at 5 hpf (coeloblastula stage; Supplementary Fig. 1c). With the onset of gastrulation at 6 hpf[30], however, all but one of the 4q blastomeres divide (Supplementary Fig. 1d). The undivided 4q cell is larger at this stage, and only cleaves into two large cells after ingression during early gastrulation at 7 hpf (Supplementary Fig. 1e, f). Therefore, the earliest morphological sign of bilateral symmetry in *O. fusiformis* occurs with the formation and deferred division of one 4q micromere, which is inferred to be the 4d cell in other conditional annelids with similar dynamics of division and ingression of the 4q blastomeres[34–37].

To dissect the patterns of ERK1/2 activity and their connection to the 4q micromeres in *O. fusiformis*, we used a cross-reactive antibody against the active di-phosphorylated form of ERK1/2 (di-P-ERK1/2)[20,24] from fertilisation to the onset of bilateral symmetry and the undivided 4q micromere (6 hpf). This annelid has a single ERK1/2 ortholog expressed at low levels in active oocytes and up to 4 hpf (~32 cell stage), when its expression and activity increases (Supplementary Fig. 2a–d). At this time point, the four most animal/apical micromeres exhibit enriched di-P-ERK1/2 signal (Fig. 2a, b; Supplementary Fig. 2c), which disappears at 5 hpf, when di-P-ERK is enriched in one 4q micromere instead (Fig. 2c, d; Supplementary Fig. 2c). At 6 hpf, di-P-ERK1/2 signal is enriched in seven blastomeres, including three pairs of micromeres of the second quartet and the 4q micromere that breaks the quadri-radial symmetry of the embryo by deferring cleavage to 7 hpf (Fig. 2e, f; Supplementary Fig. 2c). Together, our data supports that the di-P-ERK1/2 enriched 4q micromere at 5 and 6 hpf in *O. fusiformis* and that defers cleavage until 7 hpf is the 4d cell (Fig. 2g), resembling the condition observed in the annelid *H. hexagonus*[20] and many gastropod molluscs[19–22,24].

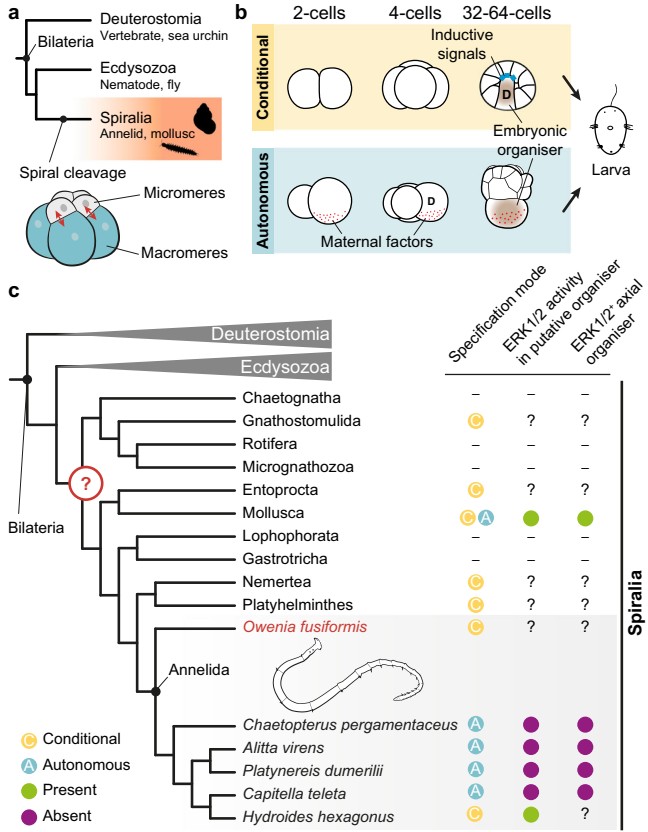

**Fig. 1 Conditional and autonomous cell fate specification occur in spiral cleavage. a** Spiral cleavage is a stereotypic developmental mode ancestral to Spiralia (e.g., snails and segmented worms), one of the three major lineages of bilaterally symmetrical animals (Bilateria). Spiral cleavage is characterised by alternating oblique cell divisions (red arrows) along the animal vegetal axis from the 4-cell stage onwards. **b** Spiral cleaving embryos specify their axial identities and embryonic organiser either conditionally or autonomously. In the conditional mode, the specification of the posterodorsal identity (D-quadrant; represented as a D) and the embryonic organiser occurs late in cleavage (~32–64 cell stages, depending on the species) through inductive signals. The autonomous mode of cell fate specification, however, relies on differentially segregated maternal factors (red dots) that specify the D-quadrant already at the 4-cell stage (cell with D). Later during cleavage, one cell of the D-quadrant that presumably inherits part of those maternal factors will act as embryonic organiser. **c** The conditional mode is found in all spiralian lineages exhibiting spiral cleavage and the autonomous mode is only observed in Mollusca and Annelida, which suggests conditional specification is ancestral. While conditional and autonomous molluscs use the MAP kinase ERK1/2 to establish their body plan, all studied autonomous annelids do not rely on this signalling cascade and knowledge outside Mollusca and Annelida is absent. The ancestral mechanisms controlling spiral cleavage are thus still unknown, which limits our understanding of the evolution of Spiralia. The conditional annelid *Owenia fusiformis*, as a member of the sister lineage to all remaining annelids, can help to infer ancestral developmental characters to Annelida and Spiralia as a whole. Drawings are not to scale.

**ERK1/2 signalling controls axial polarity.** To further examine the role of the ERK1/2 signalling during *O. fusiformis* development, we treated embryos with brefeldin A (BFA), an inhibitor of intracellular protein trafficking previously used to block the induction of the organiser in other spiral cleaving embryos;[29,38] and U0126, a selective inhibitor of MEK1/2 and ERK1/2 di-phosphorylation[24,39] (Fig. 3a). For both drugs, treatment from fertilisation (~0.5 hpf) to 5 hpf, when di-P-ERK1/2 is enriched in

4d, effectively blocks activation of ERK1/2 (Fig. 3b; Supplementary Fig. 3a; Supplementary Table 2) and causes the loss of bilateral symmetry, posterior structures (e.g., chaetae and hindgut) and larval muscles in a dosage-dependent manner up to 100% of the embryos at a 10 μM concentration (Fig. 3c; Supplementary Fig. 3b; Supplementary Tables 3 and 4). Compared to control samples, 0.5–5 hpf treated embryos lack a fully formed apical tuft and apical organ, showing reduced ectodermal expression of the apical organ marker *six3/6* and fewer apical cells positive for the neuronal marker *synaptotagmin-1* (*syt1*)[30] (Fig. 3d). In addition, treated embryos lack expression of hindgut (*cdx*) and trunk mesodermal (*twist*) markers[3], exhibit expanded expression of the oral ectodermal marker gene *gsc*[3] around the single gut opening (Fig. 3d; Supplementary Fig. 3c), and retain expression of the midgut endodermal marker *GATA4/5/6b*[3] (Supplementary Fig. 3c). We deem this phenotype as anteroventrally radialised (or Radial; Supplementary Table 3). Therefore, activation of ERK1/2 signalling in the 4d cell at the coeloblastula stage is required to specify and develop posterior and dorsal structures during *O. fusiformis* embryogenesis, although inhibition of ERK1/2 activity in the 1q111 (at 4 hpf) might also contribute to the reduced apical organ phenotype.

To dissect the exact timing of induction and activity of ERK1/2 during *O. fusiformis* cleavage, we treated embryos with 10 μM BFA/U0126 in overlapping time windows from fertilisation to early gastrulation (Fig. 3e, f; Supplementary Table 5). Blocking protein secretion with BFA from fertilisation to the 8-cell stage does not affect normal development. However, BFA treatment between the 8-cell stage and 4 hpf results in larvae with all morphological landmarks of a typical mitraria larva and normal expression of tissue-specific markers, but with a compressed morphology along the apical-ventral axis, perhaps due to an impaired formation of the larval blastocele (Fig. 3g; Supplementary Fig. 3c). Only treatment with BFA from 4–6 hpf, and hence spanning the formation of 4d, causes a radial phenotype, with treatments after 5 hpf being lethal (Fig. 3e, f; Supplementary Fig. 3c). All of this suggests that a potential inductive event driving the activation of ERK1/2 in 4d should happen between 4 and 5 hpf, right during 4q micromere formation (Supplementary Fig. 1a–c). Unexpectedly, preventing ERK1/2 di-phosphorylation with U0126 from the 2-cell stage until 4 hpf, when this induction event might begin, also causes a radial phenotype (Fig. 3e), with just slight differences between certain time points (Supplementary Fig. 2c). Therefore, ERK1/2 activity is essential for normal embryonic patterning and posterodorsal development throughout most spiral cleavage in *O. fusiformis*. However, the combination of BFA and U0126 phenotypes suggests that ERK1/2 might act autonomously from the 2-cell stage to 4 hpf, while it requires of intracellular protein trafficking—and hence potentially of inductive cell-to-cell communication signals—for its enrichment in 4d.

**ERK1/2 activates posterodorsal and mesodermal genes.** To investigate the mechanisms through which ERK1/2 controls posterodorsal development in *O. fusiformis*, we next hypothesised that genome-wide profiling of gene expression in BFA and U0126 treated embryos would uncover upstream regulators and downstream targets of ERK1/2 activity. We thus treated embryos with either 10 μM BFA or 10 μM U0126 from fertilisation to 5 hpf (to cover both the autonomous and conditional phases of ERK1/2 activity) and performed RNA-seq transcriptome profiling in treated and controlled embryos collected right after di-P-ERK1/2 enrichment in the 4d cell (coeloblastula; 5.5 hpf) and at the larval stage (Fig. 4a; Supplementary Fig. 4a, b). Differential expression analyses revealed 90 and 268 differentially expressed genes (DEGs; log$_2$ (fold change) < −1.5 and false discovery rate-adjusted

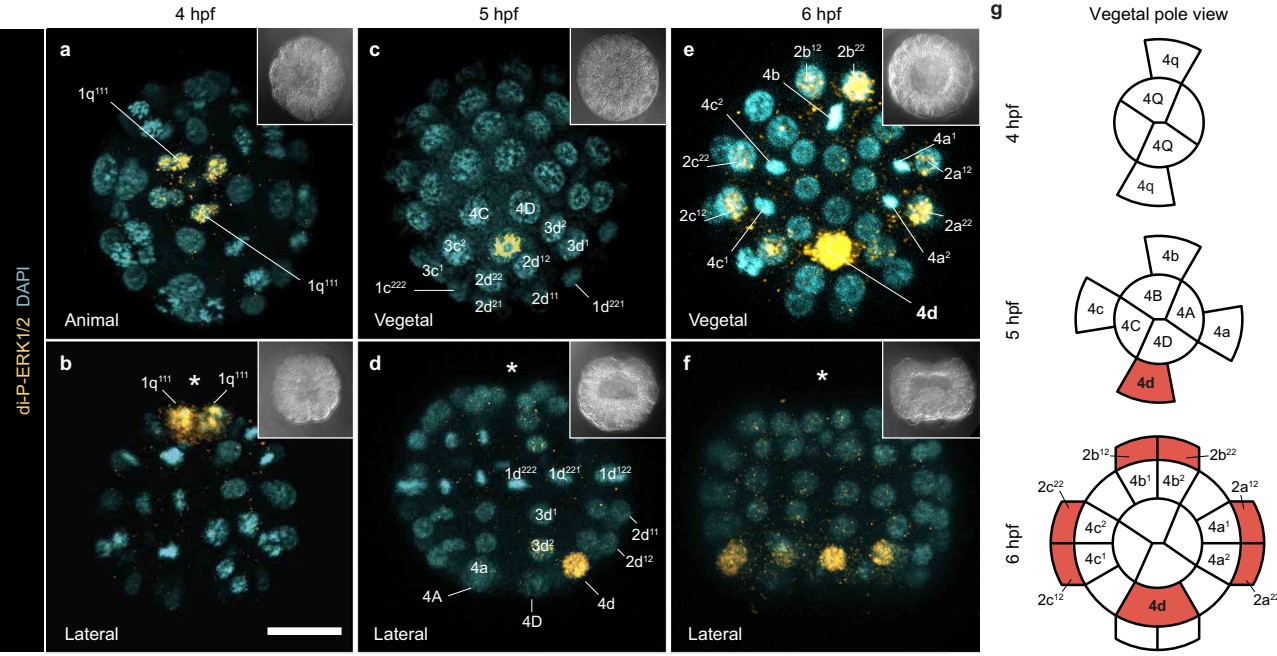

**Fig. 2 Di-phosphorylated ERK1/2 is enriched in the 4d micromere in *O. fusiformis*. a–f** z-stack projections of whole-mount embryos at 4-, 5-, and 6-h post fertilisation (hpf) stained against active di-phosporylated-ERK1/2 (di-P-ERK1/2; yellow) and nuclei (DAPI). At 4 hpf (**a**, **b**), di-P-ERK1/2 is enriched in four animal cells (1q[111]). At 5 hpf (**c**, **d**), di-P-ERK1/2 signal is in a single cell (the 4d micromere; see Supplementary Fig. 1), while active ERK1/2 occurs in six cells of the 2a-c lineage and the 4d cell at 6 hpf (**e**, **f**). Insets in **a**–**f** are bright-field images of the corresponding stacks. **g** Schematic representation of the vegetal pole during birth and first division of the fourth micromere quartet in *O. fusiformis*. In red, enriched di-P-ERK1/2 signal. In **d** and **f**, the asterisks point to the animal pole. Descriptions for **a**–**f** are based on at least ten embryos per stage, from a minimum of two biological replicates. Scale bars are 50 μm. Drawings in **g** are not to scale.

$p$-value < 0.05) in BFA-treated coeloblastulae and larvae, respectively, and 132 (coeloblastula) and 373 (larva) DEGs after U0126 treatment (Fig. 4b). When considering all comparisons and removing redundancies, we detected a total of 628 DEGs, 414 (65.92%) of which were functionally annotated and enriched in gene ontology terms related to regulation of transcription, development, and cell fate specification (Supplementary Data 2). Most of these DEGs were downregulated (Fig. 4b, c; Supplementary Fig. 4c, d) and only three DEGs (*cdx*, *fer3* and *foxH*) appeared systematically downregulated in both drugs and in the two developmental time points (Fig. 4b; Supplementary Fig. 4d). Notably, U0126 and BFA downstream targets showed limited overlap (Supplementary Fig. 4d), probably a result of the different specificity of the two treatments, with BFA broadly targeting intracellular trafficking and proteins secretion and hence potentially showing more off-target effects than U0126. Nevertheless, our approach revealed a confident and relatively small set of genes whose expression is strongly dependent on ERK1/2 activity.

To validate that the DEGs are affected by 4d misspecification, we selected 22 DEGs for further gene expression analyses (Fig. 4d; Supplementary Table 6), including a variety of transcription factors that are markers of particular cells and tissue types (e.g., *six3/6*, *gsc*, *cdx*, *AP2*, *foxQ2*), required for mesoderm development (e.g., *twist*, *hand2*, *foxH*) and neurogenesis (e.g., *POU4*, *irxA*), Wnt ligands (*wnt1*, *wntA* and *wnt4*), TGF-β modulators (*noggin* and *BAMBI*), and Notch signalling components (*delta* and *notch-like*). Stage-specific RNA-seq data covering 12 developmental time-points[33], from the unfertilized oocyte to the mature larva, confirmed that the expression of all candidate genes upregulates at the time of or just after 4d specification and di-P-ERK1/2 enrichment in this cell (Fig. 4d). These genes are expressed either in apical/anterior domains (*noggin*, *BAMBI*, *foxQ2*, *POU4*, *six3/6*, *gsc*), the posterior larval tip and chaetae (*fer3*, *lhx1/5*, *wnt1*, *notch*,

*msx2-a*, *irxA*, *AP2*, *wnt4*, *delta*), the hindgut (*cdx*), or mesodermal derivatives (*foxH*, *rhox*, *wntA*, *POU3*, *hand2*, *twist*) (Fig. 4d; Supplementary Fig. 5a–g). Analysis of the expression of these genes in control and treated embryos at the coeloblastula (5.5 hpf) and larva (24 hpf) stages confirmed the expression domains of these genes disappear after treatment with either BFA or U0126 (Fig. 4e; Supplementary Fig. 5a–g; Supplementary Data 1; Supplementary Table 7), thus validating our RNA-seq approach. Altogether, our findings uncover a set of co-regulated genes that act downstream of ERK1/2 signalling, and that are involved in the development of apical, posterodorsal and mesodermal structures in *O. fusiformis*.

**ERK1/2 signalling specifies and patterns the D-quadrant.** Our RNA-seq study and candidate gene screening revealed nine genes expressed at the vegetal pole at 5.5 hpf and whose expression was affected by either direct inhibition of ERK1/2 di-phosphorylation (*cdx*, *delta*, *foxH*, *wnt1*, *wntA*, *rhox*, *fer3* and *AP2*) or impairing intracellular protein trafficking and secretion (*gsc*) (Fig. 4e; Supplementary Fig. 5a, c, e). None of these genes are expressed at 5 hpf, at the onset of ERK1/2 activity in the 4d micromere (Fig. 5a). Instead, the early endodermal marker *GATA4/5/6a*[3], whose expression is unaffected by BFA and U0126 treatment, is symmetrically expressed in the gastral plate (including 4d) at this stage (Fig. 5a; Supplementary Fig. 6a). At 5.5 hpf, half an hour after the initial activation of ERK1/2 signalling in 4d, the vegetal pole becomes bilaterally symmetrically patterned, but only two of the ERK1/2 dependent genes (*cdx* and *delta*) are expressed in the 4d micromere (Fig. 5b, c). Notably, these are expressed in the 4d cell in other spiralian embryos (Supplementary Table 8), thus supporting our cell lineage inference based on cell cycle progression, cell positional information, and ERK1/2 activity.

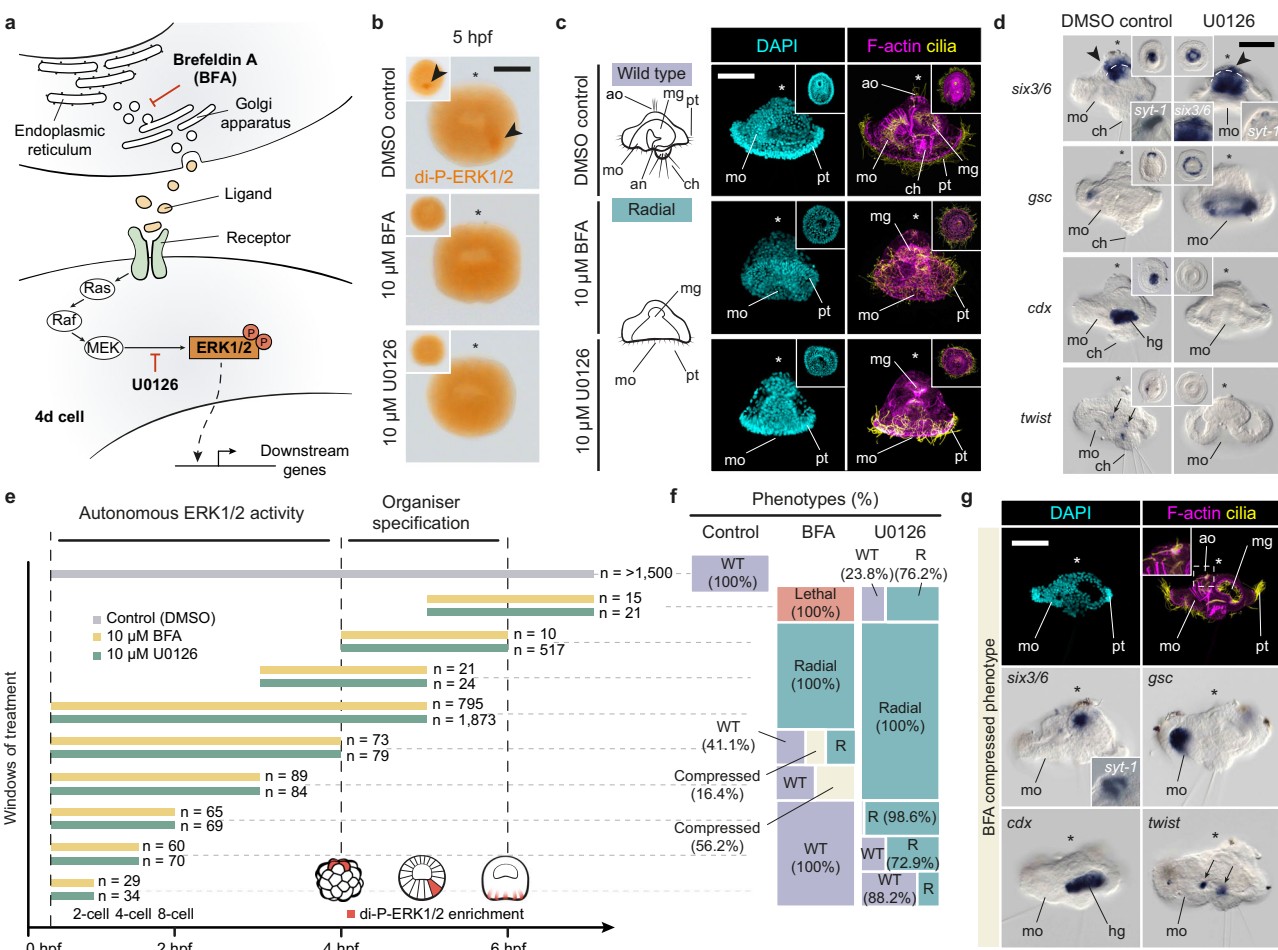

**Fig. 3 ERK1/2 signalling controls axial polarity in *O. fusiformis*. a** Schematic drawing of the ERK1/2 signalling cascade and mode of action of Brefeldin A (BFA) and U0126. **b** Whole mount immunostaining against di-P-ERK1/2 (main panels are lateral views; insets are vegetal views). BFA and U0126 treatments from 0.5 h post fertilisation (hpf) to 5 hpf prevent di-P-ERK1/2 enrichment in the 4d cell. **c** Lateral z-stack projections of control and BFA/U0126 24 hpf larvae treated from 0.5–5 hpf. Insets in treated embryos are ventral views. Cilia are labelled with anti-acetylated tubulin and F-actin with phalloidin. **d** Whole mount in situ hybridisation on control and U0126 treated (0.5–5 hpf) larvae. **e** Schematic representation of the analysed drug treatment windows. Number of cases is shown to the right of each bar specifying the time interval. **f** Distribution and percentages of the larval phenotypes observed for each window and experimental condition (WT, wild type; R, radial). **g** Phenotypic characterisation of the compressed larval phenotype obtained after BFA treatment from 0.5 hpf until 3–4 hpf. First row, lateral z-stack projections of larvae stained with DAPI (nuclei), phalloidin (F-actin) and Acetylated Tubulin (cilia). Bottom rows, lateral views of whole mount in situ hybridisation against apical ectodermal (*six3/6*), oral ectodermal (*gsc*), hindgut (*cdx*) and mesodermal (*twist*) markers. Compressed larvae are normal but have a reduced apical organ and their larval blastocele is somehow obliterated. In **b**–**d** and **g**, asterisks point to the apical pole. For **b**, Supplementary Table 2 reports detailed numbers. For **c**–**d** and **g**, Supplementary Data 1 reports detailed numbers. For **e** and **f**, Supplementary Table 5 reports detailed numbers. Schematic drawings are not to scale. ao apical organ, an anus, ch chaetae, mg midgut, mo mouth, pt prototroch. Scale bars are 50 μm.

The ParaHox gene *cdx*, which is detected in the hindgut of *O. fusiformis*[3] (Fig. 4e; Supplementary Fig. 6b), becomes expressed in 4d, and later on in the two daughter cells of 4d (i.e., the left and right mesentoblasts, or ML and MR cells), which remain undivided and continue exhibiting active di-P-ERK1/2 until the gastrula stage (Fig. 6a, b; Supplementary Fig. 1f). Similarly, the Notch-ligand *delta* (Supplementary Fig. 6c, d) is expressed in 4d at 5.5 hpf, but also in most of the descendants of 1d at the animal pole, plus animal micromeres and ectodermal derivatives of the C- and D-quadrant at the vegetal pole (Fig. 5b, c; Supplementary Fig. 5a). To investigate how ERK1/2 might control activation of *cdx* and *delta*, we used Assay for Transposase-Accessible Chromatin using sequencing (ATAC-seq) data at 5 hpf to identify transcription factor motifs present in accessible chromatin regions associated with these two genes (Fig. 5d). These include motifs of transcriptional regulators known to be modulated by ERK1/2 phosphorylation, such

as ETS, RUNX and GATA factors[18]. Therefore, ERK1/2 di-phosphorylation in 4d seems to control the activity of transcriptional regulators that induce posterior fates (*cdx*) and cell–cell communication genes (*delta*).

The other six additional ERK1/2-dependent genes pattern the micromeres surrounding 4d at 5.5 hpf, defining mesodermal and posterior ectodermal domains (Fig. 5b, c). The transcription factor *foxH* (Supplementary Fig. 6e), which regulates mesoderm development during gastrulation in vertebrate embryos[40–43], is detected in four micromeres adjacent to 4d (Fig. 5b, c), which might contribute to lateral ectomesoderm according to lineage tracing studies in the annelid *Urechis caupo*[37]. During gastrulation, these *foxH* + cells undergo asymmetric cell division at orthogonal cleavage planes to end up surrounding the MR and ML cells (Fig. 6d, e). Later during axial elongation, *foxH* expression forms a posterior V-shaped pattern of putative mesodermal precursors, fading away in larval stages (Supplementary Fig. 6b). The ligands

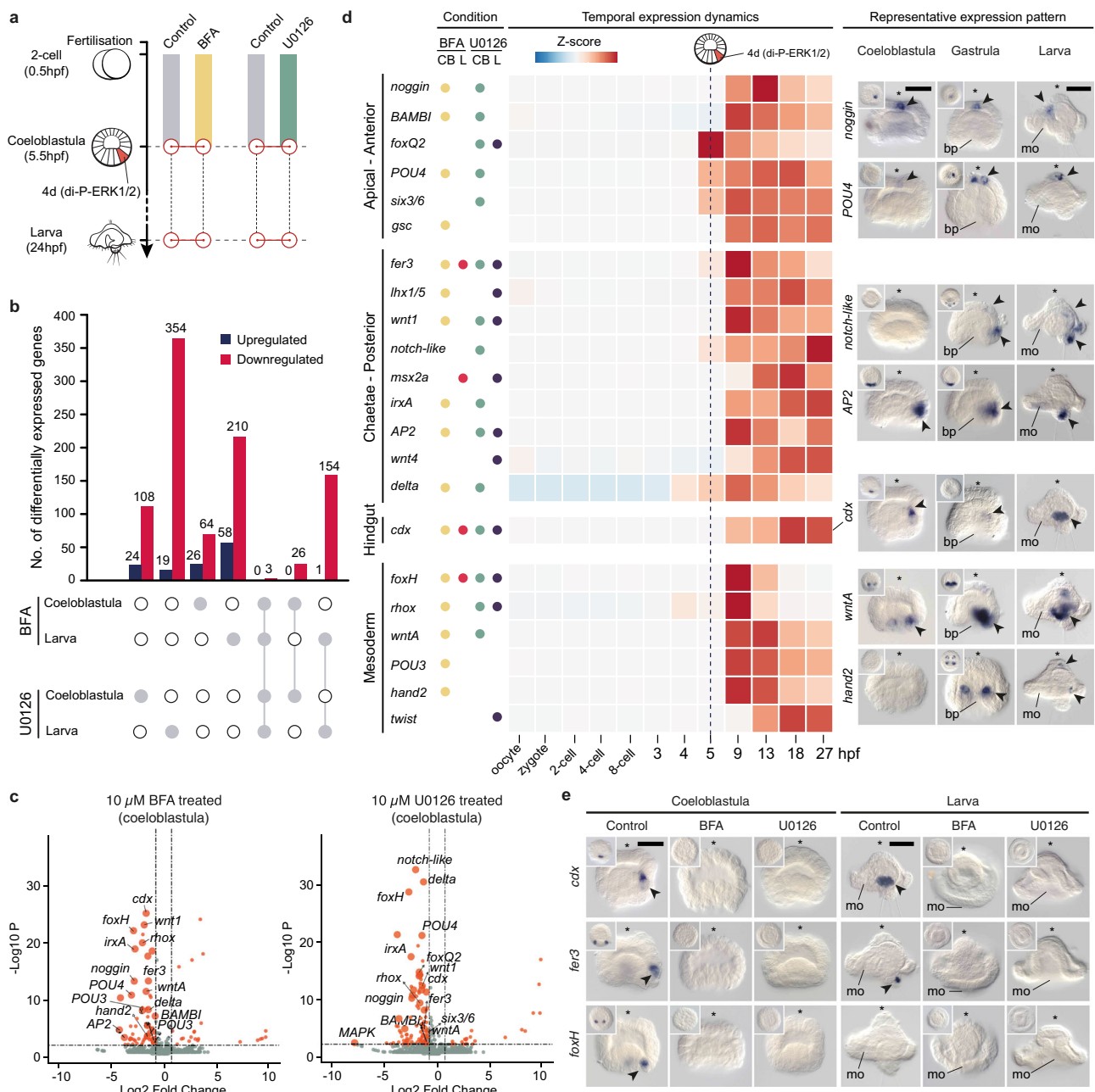

**Fig. 4 ERK1/2 signalling activates posterodorsal and mesodermal genes. a** Experimental design for RNA-seq collections. Solid colour bars show period of control (DMSO), brefeldin A (BFA) and U0126 treatments. Red circles indicate sample collection at the coeloblastula (5.5 h post fertilisation, hpf) and larval stage (24 hpf). **b** Number of differentially expressed (DE) genes in different conditions and comparisons (indicated by white and grey dots in the bottom; grey dots highlight conditions included in the comparison), where red bars are downregulated genes and blue bars are upregulated genes. **c** Volcano plots (two-sided Wald test) for BFA and U0126 treated coeloblastula. Red dots show DE genes, with candidate genes labelled for each comparison. **d** Temporal and spatial time course of expression of candidate genes affected by 4d misspecification, which are generally associated with apical/anterior, chaetae/posterior, hindgut, and mesoderm development. The first column ("Condition") indicates the treatment (BFA or U0126) and developmental stage (CB coeloblastula, L larva) at which each of the candidate genes is differentially expressed (each condition highlighted with a different colour). The central heatmap shows normalised z-score values of expression for each gene. The vertical dotted line highlights the timing of 4d specification. The third column shows whole mount in situ hybridisation images of a subset of these 22 candidate genes at the coeloblastula (5 hpf), gastrula (9 hpf) and larval stages (24 hpf). **e** Only three of the candidate genes (cdx, foxH and fer3) are downregulated in all stages and treatment comparisons. Validation via in situ hybridisation demonstrates the expression of these genes is lost after drug treatment. In **d** and **e**, the pictures for cdx at the coeloblastula and larval stage, control condition, are the same. In **d** and **e**, asterisks point to the animal/apical pole, and arrowheads to the domains of expression. In **d** heatmaps were generated from a developmental time course generated from two biological replicates, and in situ expression was done with a minimum of two biological replicates. For **c** and **e**, Supplementary Data 1 reports detailed numbers. Scale bars are 50 μm. Schematic drawings are not to scale. bp blastopore, mo mouth.

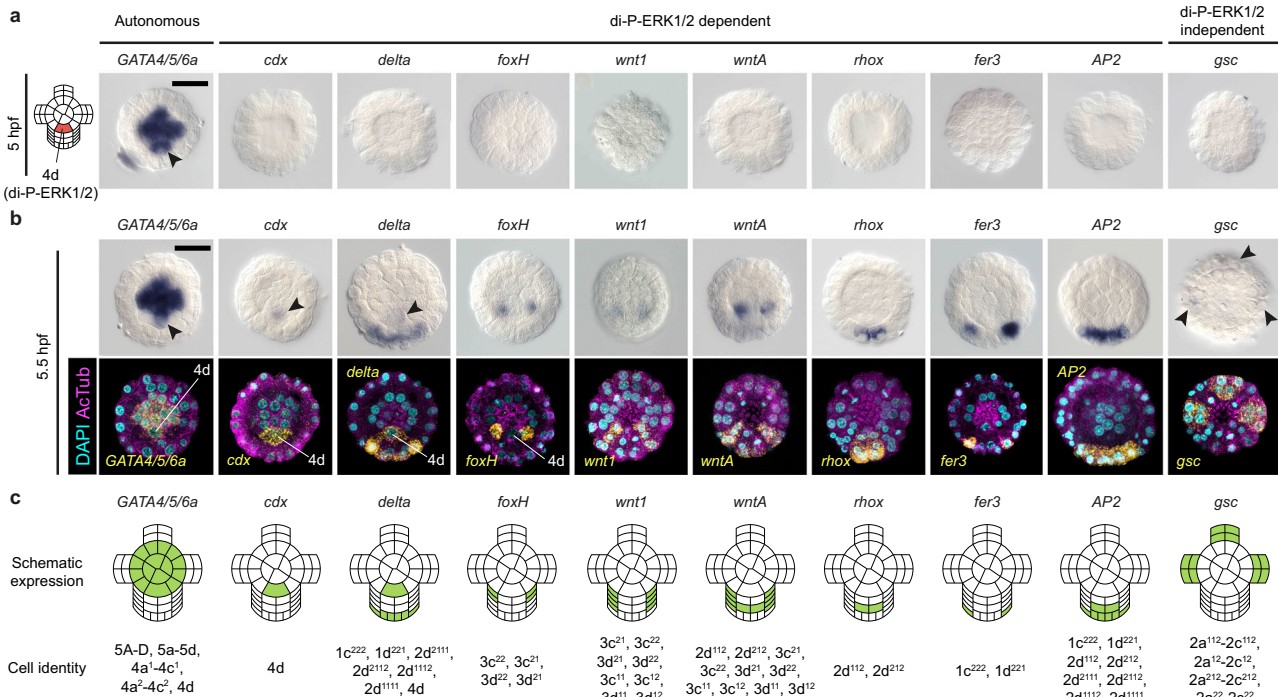

**Fig. 5 ERK1/2 signalling specifies and patterns the D-quadrant. a, b** Whole mount in situ hybridisation of a subset of genes (ERK1/2-dependent and independent) patterning the vegetal pole at 5- and 5.5-h post fertilisation (hpf) respectively. At 5 hpf (**a**), when the 4d is specified (left drawing) only the endodermal marker *GATA4/5/6a* is expressed (including in the 4d cell, arrowhead). At 5.5 hpf (**b**), *cdx* and *delta* become expressed in the 4d cell (arrowheads) and the C- and D-quadrant micromeres surrounding the 4d cell start expressing a range of genes involved in posterodorsal and mesodermal development (*foxH, wnt1, wntA, rhox, fer3* and *AP2*). The anterior oral ectodermal gene *gsc*, which is not directly regulated by ERK1/2 activation, also starts to be expressed at this time point (arrowheads). In **b**, the upper raw are colorimetric in situ hybridisations and the lower row are z-stack projections of fluorescent in situ hybridisations counterstained for nuclei (with DAPI) and cell borders (with Acetylated Tubulin immunohistochemistry). In **a** and **b**, all panels are vegetal views. Descriptions for **a** and **b** are based on at least ten embryos per stage, from a minimum of two biological replicates. **c** Schematic representation of the vegetal pole and micromeres of the C- and D-quadrant around the 4d blastomere depicting gene expression domains (in green) and inferred cell identities of the studied genes at 5.5 hpf. Scale bars are 50 μm. Drawings are not to scale.

*wnt1* and *wntA* (Supplementary Fig. 6f), expressed in the posterior region and D-quadrant during development in the annelid *Platynereis dumerilii*[44], are expressed in two bilaterally symmetrical columns of micromeres and *wntA* is also detected in two additional D-quadrant micromeres (Fig. 5b, c). The homeobox *rhox* (Supplementary Fig. 6g), which is expressed in male and female primordial germ cells in vertebrates[45], is detected in two vegetal micromeres in the D-quadrant (Fig. 5b, c) and thereafter in two small cells inside the blastocele in the gastrula (Supplementary Fig. 5e). The transcription factors *fer3* and *AP2* are expressed in two single micromeres and in a broader posterior ectodermal domain, respectively, becoming restricted to a small expression domain at or near the posterior chaetal sac of the larva (Fig. 5b, c; Supplementary Fig. 5c and 6b). Notably, the 4d cell, which has a larger cell size than the other 4q micromeres because it does not divide at 5.5 and 6 hpf (Supplementary Fig. 1), establishes almost as double direct cell–cell contacts with its surrounding cells—including most of the cells expressing *foxH, wnt1, wntA, rhox,* and *AP2*—than each of the daughter cells of 4a–c (Fig. 6e). Therefore, these results support a probable inductive role of the 4d in defining mesodermal and posterodorsal fates, although cell ablation and/or cell transplantation studies are required to experimentally validate this hypothesis.

The homeobox gene *gsc* is the only candidate expressed outside the D-quadrant at 5.5 hpf, being detected in a U-shape domain of micromeres of the A, B and C quadrants that occupy the prospective anterolateral blastoporal rim (Fig. 5b, c). The expression of *gsc* is independent of ERK1/2 activity, consistent with its location outside the D-quadrant, but is downregulated after BFA treatment (Supplementary Data 1). Accordingly, *gsc*

expression disappears in BFA-treated coeloblastulae, while inhibition of ERK1/2 activity with U0126 expands *gsc* domains, which becomes detected in all embryonic quadrants in a radial fashion (Fig. 6f; Supplementary Table 9). Indeed, in ERK1/2 inhibited embryos all 4q micromeres cleave into $4q^1$ and $4q^2$, as no 4q becomes the 4d cell (Fig. 6f). Therefore, all quadrants adopt anteroventral fates (*gsc*), preventing the expression of posterior marker genes such as *AP2* (Fig. 6f; Supplementary Table 9). These results demonstrate that the radial phenotype observed after U0126 is a consequence of misspecifying the 4d cell, and not a result of the D-quadrant becoming specified but not developing further. In addition, this data also demonstrates that specification of the 4d cell through ERK1/2 activity represses anterior fates, as shown by limiting *gsc* expression.

**Notch signalling regulates posterodorsal development.** The upregulation of the Notch-ligand *delta* in 4d after ERK1/2 activation suggests that the role of 4d in promoting posterodorsal development might occur by direct cell–cell communication mediated by the Notch signalling pathway. To test this hypothesis, we treated embryos before and after the specification of 4d with the small molecule LY411575, which is a selective inhibitor of the gamma-secretase that cleaves the active Notch intracellular domain upon Notch activation by Delta[46] (Fig. 7a, b). Larvae growing from embryos treated with LY411575 from fertilisation to 5 hpf exhibit a normal phenotype, with an obvious bilateral symmetry and normal organogenesis (Fig. 7b–e; Supplementary Table 10). However, inhibition of the Notch pathway from 5 hpf,

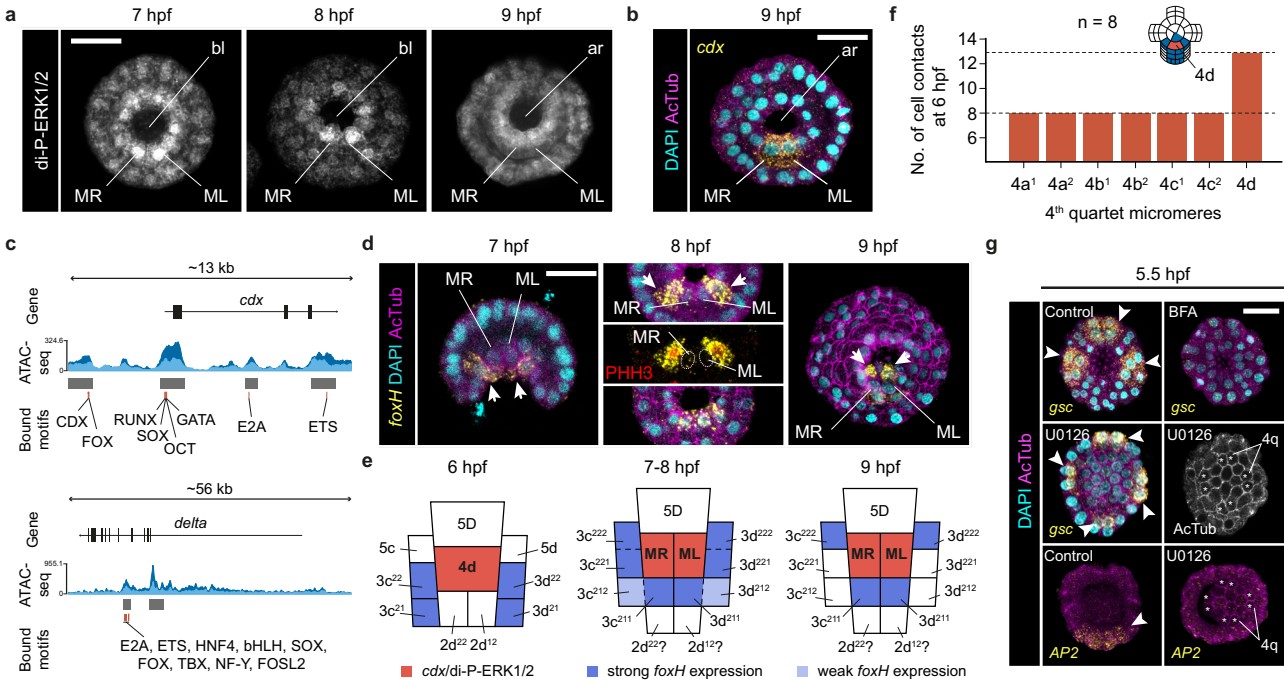

**Fig. 6 The behaviour of 4d during gastrulation. a** z-stack projections (vegetal views) of whole mount immunohistochemistry against di-phosphorylated ERK1/2 (di-P-ERK1/2) from 7 to 9 h post fertilisation (hpf). The 4d micromere cleaves at 7 hpf into the MR and ML daughter cells (Supplementary Fig. 1). **b** z-stack projection (vegetal view) of a fluorescent whole mount in situ hybridisation against *cdx* (yellow) at the gastrula stage (9 hpf), counterstained with DAPI (nuclei) and acetylated tubulin (cell boundaries). **c** Distribution of transcription factor motifs accessible in open chromatin regions around the *cdx* (upper part) and *delta* (lower part) loci at 5 hpf. Dark and light blue depict the two ATAC-seq replicates. **d** z-stack projections (vegetal views) of fluorescent whole mount in situ hybridisations against *foxH* from 6 to 9 hpf (i.e., gastrulation), counterstained with DAPI (nuclei), acetylated tubulin (cell boundaries) and phospho-Histone3 (red; marker of mitosis). **e** Schematic drawings of the cells surrounding 4d and MR/ML during gastrulation, depicting the expression of *foxH* based on **d**. **f** Number of cellular contacts for each of the 4q micromeres at 5.5 hpf based on the analysis of eight embryos stained for membrane actin. The 4d cell establishes more cell contacts than any other 4q micromere at the onset of gastrulation. **g** z-stack projections (vegetal views) of fluorescent whole mount in situ hybridisations against *gsc* and *AP2* at 5.5 hpf, counterstained with DAPI (nuclei) and acetylated tubulin (cell boundaries). 4d misspecification expands the oral ectodermal gene *gsc* and impairs the expression of the posterior gene *AP2*, as all 4q micromeres (asterisks) behave similarly. *gsc* expression is lost in brefeldin A (BFA) treated embryos, which suggests it might require inductive signals. In **d** and **g**, arrows and arrowheads point to expression domains. Descriptions for **a**, **b** and **d** are based on at least ten embryos per stage, from a minimum of two biological replicates. For **g**, Supplementary Table 9 reports detailed numbers. Scale bars are 50 μm. Schematic drawings are not to scale. ar archenteron, bl blastopore.

when ERK1/2 activity specifies the 4d cell, onwards results in larvae with bilateral symmetry, a through gut with a fully anteriorised mouth, an apical organ, but shorter posterodorsal tissues, a phenotype we call 'stubby' (Fig. 7b, c, f; Supplementary Table 10). Accordingly, these larvae express *gsc* in the anterior oral ectoderm and have a short hindgut expressing *cdx* (Fig. 7g; Supplementary Table 11). The 'stubby' phenotype thus differs from the radial one observed when inhibiting ERK1/2 in that axial identities are specified in the former, but only anteroventral fates in the later (Supplementary Table 3). Notably, the posterodorsal ectoderm of the gastrula expresses a *notch-like* ligand and as observed for *delta*, its expression is regulated by ERK1/2 activity (Fig. 4d; Supplementary Fig. 5f). Together, our findings support that activation of ERK1/2 in the 4d cell facilitates Notch signalling in this and surrounding posterodorsal cells, which is altogether necessary for normal development of posterior and dorsal structures. Further studies will, however, elucidate the exact gene regulatory network connecting ERK1/2 and Notch signalling to control axial development in *O. fusiformis*.

**FGF signalling regulates ERK1/2 di-phosphorylation in 4d.** In vertebrate embryos, FGF signalling mediated by ERK1/2 activity regulates the expression of *cdx* and *delta* in the developing hindgut and presomitic mesoderm during posterior trunk elongation[47]. We

thus hypothesised that FGF signalling might be the upstream regulator driving ERK1/2 activity in 4d in *O. fusiformis* (Fig. 8a), which is also required to induce the expression of *cdx* and *delta* and specify the hindgut and trunk mesodermal progenitor. *Owenia fusiformis* has a single FGF receptor (FGFR; Supplementary Fig. 7a) with high amino acid conservation at key functional residues compared to the human FGFR1 ortholog (Supplementary Fig. 7b). While it is transcriptionally upregulated at the gastrula stage and appears to be expressed mostly in mesodermal derivatives from gastrula onwards (Supplementary Fig. 7c, d), *FGFR* appears weakly expressed in the gastral plate at the coeloblastula stage in *O. fusiformis* (Fig. 8b), as also observed in brachiopods and phoronids[48]. At this stage, FGF ligands are only weakly expressed (Supplementary Fig. 7c) and not detected by in situ hybridisation. Treatment with 30 μM SU5402, a selective inhibitor of FGFR phosphorylation and activation[49] (Fig. 8a), prevents di-P-ERK1/2 enrichment at 4d cell in 96% of the treated embryos at 5 hpf (Fig. 8c; Supplementary Fig. 7e). Indeed, treatment with SU5402 during the specification of the 4d micromere (4–6 hpf and 3–7 hpf; Fig. 6c; Supplementary Table 12) results in an anteriorly radialised phenotype, with embryos developing into larvae lacking posterior (hindgut and *cdx* expression) and reduced mesodermal (muscles and *twist* expression) structures and showing radially expended oral ectodermal fates (*gsc*) (Fig. 8g; Supplementary Table 3). Therefore, from a morphologically and gene marker perspective,

this radial phenotype is like the one obtained after inhibiting ERK1/2 di-phosphorylation with U0126 and BFA. Conversely, treatment before di-P-ERK1/2 enrichment in 4d (3–5 hpf) causes a slightly compressed phenotype (Supplementary Fig. 6f), similar as well to the one observed when blocking intracellular protein trafficking with BFA at equivalent time points. Therefore, inhibition of FGFR blocks ERK1/2 activation and phenocopies the effect of U0126 and BFA (Fig. 3c), suggesting that FGF signalling activity is upstream of and necessary for 4d specification in *O. fusiformis*.

## Discussion

Combining functional characterisation of the FGFR, ERK1/2 and Notch signalling pathways with comparative transcriptomics in *O. fusiformis*, an annelid with conditional mode of development, our study solves long-standing questions[8] on the ancestral genetic principles controlling early body patterning during spiral cleavage. Differing from what is observed in autonomous annelids[25–28], our study provides compelling evidence that FGFR and ERK1/2 signalling induce axial patterning by specifying the 4d micromere—the putative embryonic organiser—and activating downstream genes involved in mesoderm and posterodorsal development in *O. fusiformis* (Fig. 9a). As observed in some molluscs[10,20], FGFR/ERK1/2 activity breaks the quatri-radial symmetry of the embryo by delaying cell cycle progression in the 4d cell (Fig. 2e; Fig. 6f; Supplementary Fig. 1c, d), and so arresting cell division, instead of promoting it[18], appears to be a general role of ERK1/2 signalling in the context of axial patterning in spiralian embryos. Concomitant with the specification of 4d and posterodorsal identities, anterior fates become restricted to the A–C quadrants (Fig. 6f), which suggests that unknown opposing signals—one specifying anterior fates and another driving 4d specification via FGFR/ERK1/2—might act simultaneously to define axial polarity in *O. fusiformis* (Fig. 9a). While further work is needed to identify these signals, we hypothesise that the anteriorising cue must come from either cell lineages already specified at those stages (e.g., the midgut and the apical organ) or from cell-autonomous maternal determinants that are selectively cleared from the D-quadrant upon 4d specification. In addition, FGFR/ERK1/2 activity upregulates the Notch-ligand *delta* in the 4d cell and a *notch-like* receptor in the posterodorsal ectoderm of the gastrula (Fig. 4d; Fig. 5b), and thus Notch-Delta signalling—a pathway broadly used to fine-tune cell fates by direct cell-to-cell inhibition/induction[50]—is one of the downstream effectors required for posterodorsal and mesodermal development in *O. fusiformis* (Fig. 7f, g; Fig. 9a). Altogether, our findings depict a comprehensive model of the early development of an annelid with conditional spiral cleavage, setting the grounds for further studies that dissect the exact regulatory logic underpinning body patterning in this animal clade and Spiralia generally.

The activation of FGFR/ERK1/2 in the 4d blastomere to control axial identities and body patterning in *O. fusiformis* is consistent with the activity of ERK1/2 in the 4d cell in the conditional annelid *H. hexagonus*[20]. It also mirrors the condition observed in molluscs, which despite somewhat diverse patterns of early ERK1/2 activity[10,21] (Supplementary Table 1), generally use active di-phosphorylated ERK1/2 to specify and regulate the posterodorsal embryonic organiser in either the 3D blastomere or its daughter 4d cell. Indeed, the contribution of 4d to hindgut in *O. fusiformis* is also observed in molluscs[51,52], but not in some autonomous annelids[53], and Notch signalling might also underpin the potential instructing role of the 4d cell in the mollusc *Tritia obsoleta*[54] (Fig. 9a). Yet the specification of the D-quadrant and embryonic organiser relies on direct cell–cell contacts—and currently unknown inductive signals—between animal micromeres and the prospective 3D cell in molluscs[55], which is

unlikely to occur in *O. fusiformis*[30] and other conditional annelids[29] with large blastocoeles at the time of formation of the 4q micromeres. Consequently, the upstream regulators of ERK1/2 activity in the 3D/4d cells might differ between molluscs and annelids, and thus there is the possibility that the similarities in the intracellular pathways and readouts during axial specification observed between these two groups are the result of convergent evolution. However, an alternative, more parsimonious scenario is that the activation and axial patterning role of ERK1/2 in the cell that acts as embryonic organiser is homologous to Annelida and Mollusca (Fig. 9b) and thereby a fundamental trait of spiral cleavage. The array of ERK1/2 patterns observed in autonomous annelids[26–28] (Supplementary Table 1) would thus represent independent losses of an ancestral ERK1/2+ organiser related perhaps to a transition to an autonomous mode of cell fate specification. In the future, a better understanding of the upstream gene regulatory networks controlling ERK1/2 activity in annelids and molluscs will help to test this scenario and dissect the mechanisms promoting variability in the cellular identity of the embryonic organiser across Spiralia.

More generally, our study uncovers striking similarities between the molecular cascade specifying the embryonic organiser and posterodorsal identities in conditional spiralians and the gene modules regulating axial patterning and posterior growth in Deuterostomia and Ecdysozoa[47,48,56–58] (Fig. 9b; Supplementary Table 13). Although the exact regulatory logics vary across major lineages—and sometimes even between phylogenetically related clades—ERK1/2 activity is central to dorsoventral patterning in echinoderms, hemichordates, and chordates, and together with FGFR contributes to posterior growth in chordates, promoting the expression of *cdx* and *delta* at the posterior end of the growing tail (Fig. 9b; Supplementary Table 14 and references therein). Similarly, FGFR signalling drives dorsoventral patterning in an arthropod, the spider *Parasteatoda tepidariorum* (Supplementary Table 14). The Notch-Delta signalling pathway is also involved in posterior elongation in hemichordates, chordates, arthropods, and controls posterior and dorsoventral identities in nematodes (Fig. 9b; Supplementary Table 14 and references therein). Therefore, ancient and broadly conserved genetic and development principles underpin the conditional mode of cell fate specification in spiral cleavage, further reinforcing the view that this mode of development represents the ancestral condition in Spiralia[11–13]. Differently from other early embryogenesis, however, spiral cleavage, with its highly unique stereotypic pattern of cell divisions and early development with a reduced number of blastomeres, combines the instruction of axial identities and posterior development temporally and spatially together in a single organising cell—the 3D/4d in molluscs and most likely the 4d cell in annelids (Fig. 9b). How this cell, and its conserved instructing patterning role promotes the subsequent development of profoundly different adult morphologies in Spiralia remains an open question, which a detailed investigation of the genetic modules downstream of the D-quadrant organiser across spiralian lineages will help to resolve.

## Methods

**Animal husbandry and embryo collections**. Adult specimens of *O. fusiformis* Delle Chiaje, 1844 were collected from the coast near the Station Biologique de Roscoff (France) during the reproductive season (May–July). In the lab, animals were kept in artificial seawater (ASW) at 15 °C. In vitro fertilizations were conducted as previously described[3,30] and embryos develop in glass bowls with filtered ASW at 19 °C until the desired embryonic stage.

**Drug treatments**. Embryos were treated with either brefeldin A (BFA; Sigma-Aldrich, #B7651), U0126 (Sigma-Aldrich, #U120), LY411575 (Sigma-Aldrich, #SML0506), or SU5402 (Sigma-Aldrich, #572630). All drug stocks were made in

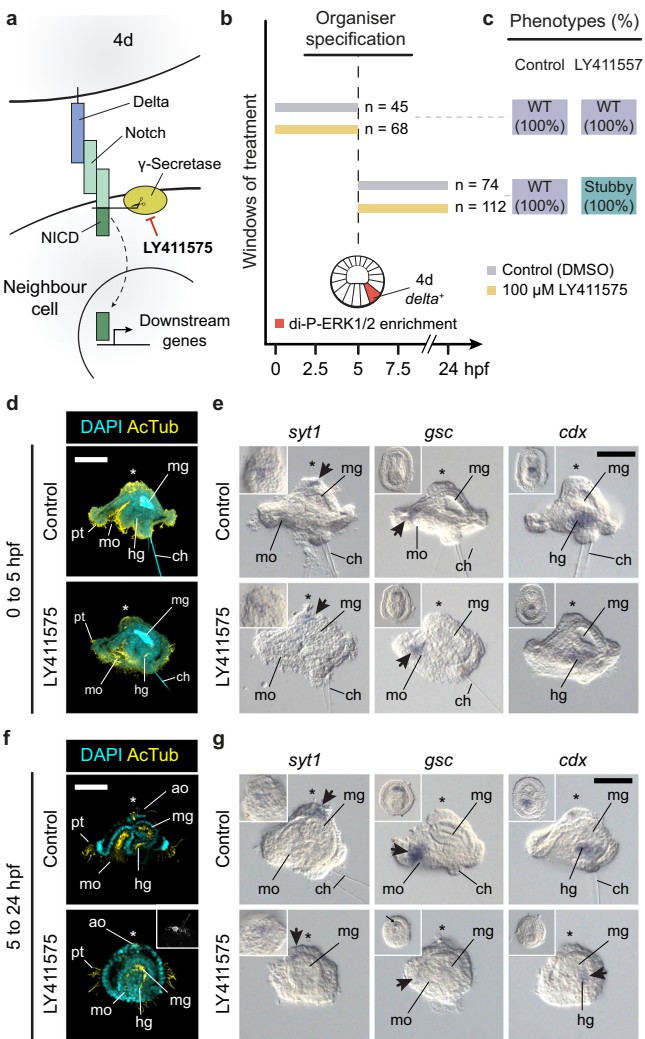

**Fig. 7 Notch signalling is required for normal posterodorsal development after D-quadrant specification. a** Schematic drawing of the Notch-Delta signalling pathway, with the Delta ligand being expressed in the 4d cell (see Fig. 5b) and a Notch receptor in its neighbouring cells (Fig. 4d). **b** Schematic representation of the two drug treatment windows conducted in this study (number of cases is shown to the right of each bar specifying the time interval). To assess the role of the Notch signalling pathway after *delta* upregulation in 4d, we treated embryos before and after its specification. **c** Distribution and percentages of the larval phenotypes observed for each window and experimental condition (WT, wild type). **d** z-stack projections of control and LY411575 treated embryos (from 0 to 5 hpf) stained with DAPI (nuclei) and acetylated tubulin (AcTub; cilia). Control and treated embryos show normal morphology. **e** Lateral views of whole mount in situ hybridisation of markers for apical neural cell types (*syt1; synaptotagmin-1*), oral ectoderm (*gsc*) and 4d-derived tissues and hindgut (*cdx*) in a treatment condition as in **d**. Control and larvae from treated embryos show normal gene expression patterns. Insets are ventral views and arrows point to expression domains. **f** z-stack projections of control and LY411575 treated embryos (from 5 to 25 hpf) stained with DAPI (nuclei) and acetylated tubulin (AcTub; cilia). Larvae from treated embryos exhibit bilateral symmetry and a normal apical organ, but have a reduced posterodorsal side, with a shorter/abnormal hindgut and underdeveloped chaetal region. We termed this phenotype "stubby". **e** Lateral views of whole mount in situ hybridisation of markers for apical neural cell types (*syt1*; synaptotagmin-1), oral ectoderm (*gsc*) and 4d-derived tissues and hindgut (*cdx*) in a treatment condition as in **f**. Insets are ventral views and arrows point to expression domains. In **d**–**g**, asterisks mark the anterior/apical pole. Descriptions for **b** are based on at least ten embryos per stage, from a minimum of two biological replicates. For **b**–**g**, Supplementary Table 11 reports detailed numbers. Scale bars are 50 μm. ch chaetae, hg hindgut, mg midgut, mo mouth, pt prototroch.

diaminobenzidine (DAB) (Vector Laboratories, #SK-4100) or a Tyramide Signal Amplification kit (Akoya Biosciences, #NEL742001KT) following manufacturer's recommendations. DAB-developed samples were stored in 70% glycerol and counterstained with 4′,6-diamidino-2-phenylindole (DAPI, ThermoFisher Scientific, #D3571, 1:1000) nuclear marker. Samples for laser scanning confocal microscopy (LSCM) were counterstained with DAPI diluted in 1% bovine serum albumin (BSA) in PTx for 1 h at RT and mounted in 70% glycerol.

dimethyl sulfoxide (DMSO) and diluted in ASW prior to use, with optimal working concentrations (10 μM for BFA and U0126, 100 μM for LY411575, and 30 μM for SU5402) established after initial titration. For all drug treatments, equivalent volumes of DMSO were used as negative control, and in all cases, treatments were initiated at ~0.5 h post fertilisation (hpf), right after fertilisation and sperm removal, or as indicated in each figure. Treatments were stopped by washing out the drug with at least three washes in ASW and embryos were either collected for downstream analyses or raised until larval stage. Samples collected for immunohistochemistry and gene expression analyses were fixed in 4% paraformaldehyde (PFA) in ASW for 1 h at room temperature (RT). Larvae were relaxed in 8% magnesium chloride (MgCl₂) prior to fixation and fixed in 4% PFA in 8% MgCl₂ for 1 h at RT. After fixation, samples were washed several times with 0.1% Tween-20 phosphate-buffered saline (PTw), and either stored in PTw at 4 °C for immunohistochemistry or dehydrated to 100% methanol and stored at −20 °C for whole mount in situ hybridisation. Treated and control embryos and larvae collected for RNA-seq analyses were snap frozen in liquid nitrogen and stored at −80 °C before total RNA extraction.

**Immunohistochemistry.** F-actin labelling, and antibody staining were conducted as previously described[30], with the modification that samples collected for anti-di-P-ERK1/2 staining were washed in PTw supplemented with 1:100 dilution of phosphatase inhibitors (Cell Signalling, #5872) after fixation. The primary antibodies mouse anti-double phosphorylated ERK1/2 (Sigma-Aldrich, #M8159, 1:200) and mouse anti-acetylated α-tubulin (clone 6-11B-1, Millipore-446 Sigma, #MABT868, 1:500) were diluted in 5% normal goat serum (NGS) in 0.1% Triton X-100 phosphate-buffered saline (PTx) and incubated overnight (ON) at 4 °C. After several washes in 1% bovine serum albumin (BSA) in PTx, samples were incubated with either an anti-mouse peroxidase (POD) conjugated antibody (Millipore-Sigma, #11207733910, 1:100) or an AlexaFluor594 conjugated antibody (ThermoFisher Scientific, #A32731, 1:600) diluted in 5% NGS in PTx ON at 4 °C. For samples incubated with mouse anti-double phosphorylated ERK1/2 and anti-mouse POD conjugated antibody, signal was developed using either

**RNA-seq profiling and differential gene expression analyses.** Total RNA was extracted using the Monarch total RNA Miniprep Kit (New England Biolabs, #T2010) and used for Illumina stranded mRNA library prep. Sequencing was performed on an Illumina HiSeq 4000 platform in 75 bases paired-end output mode, generating ~20 million reads per sample. Adapter sequences and low-quality bases were removed using Trimmomatic v.0.36[59]. Cleaned reads were then mapped to *O. fusiformis* reference genome annotation[33] (available at the European Nucleotide Archive repository under accession number GCA_903813345) using Kallisto v.0.44.0 to generate gene expression counts[60]. Differential gene expression analyses were performed independently for each drug and computed in pairwise comparisons of different stages and conditions using the R package DEseq2 v.1.38.0[61]. Significance threshold was adjusted to a *p*-value ≤ 0.05 and a Log2 fold change >1.5 or <−1.5 (Supplementary Data 1). PCAs and hierarchical clustering were plotted using ggplot2 and pheatmap R packages, respectively. Volcano plots were obtained with the package EnhancedVolcano, available in R. Candidate genes for further gene expression analyses were selected based on being differentially expressed in both drug treatments (U0126 and BFA) or highly differentially expressed at 5.5 hpf in U0126 treated embryos. We further refined our selection based on Gene Ontology (GO) categories (Supplementary Data 2), focusing on transcription factors and developmental genes (Supplementary Table 6).

**GO and KEGG enrichment analyses.** For GO mapping, the GO terms of differentially expressed genes associated with each comparison were extracted from *O. fusiformis* reference genome annotation[33] (available at the repository https://github.com/ChemaMD/OweniaGenome). GO enrichment analyses were implemented using the GOseq R package, correcting for gene length bias[62], and analysing upregulated and downregulated genes independently. GO terms with corrected *p*-values < 0.05 were deemed significantly enriched (Supplementary Data 2). A similar approach was followed for Kyoto Encyclopaedia of Genes and Genomes (KEGG) pathway enrichment analyses, mapping differentially represented KO numbers to their respective KEGG pathway (Supplementary Data 1).

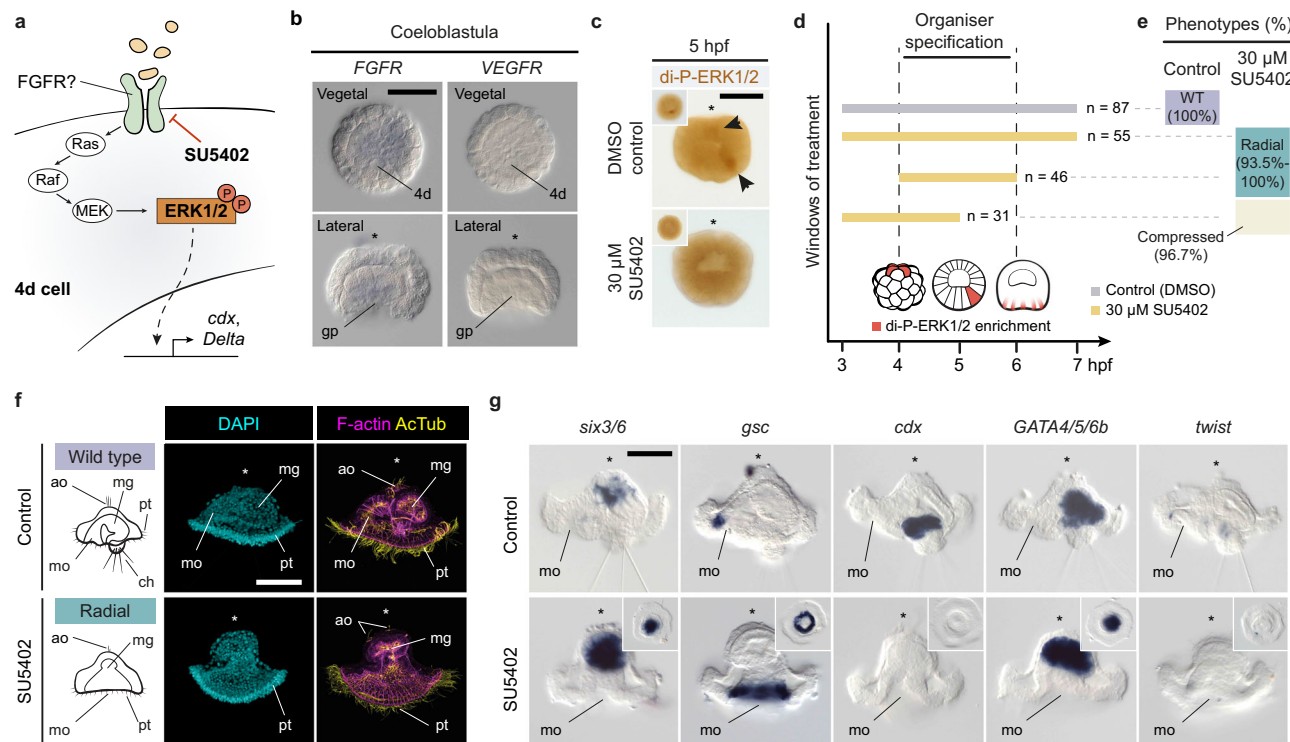

**Fig. 8 FGFR signalling regulates ERK1/2 di-phosphorylation in 4d. a** Schematic drawing of the downstream intracellular FGFR pathway. The drug SU5402 specifically inhibits receptors tyrosine kinase of the type FGF/VEGF, which can signal through the ERK1/2 signalling cascade. **b** Whole mount in situ hybridisation for FGFR and VEGFR at the coeloblastula stage in *O. fusiformis*. FGFR is expressed weakly at the gastral plate at the time of 4d-instructed posterior re-patterning. However, VEGFR, which SU5402 treatment can also inhibit, is not expressed at this stage. **c** Whole mount immunohistochemistry against di-phosphorylated ERK1/2 (di-P-ERK1/2). SU5402 treated embryos lose ERK1/2 enrichment in a vegetal micromere at 5 h post fertilisation (hpf). Main images are lateral views and insets are vegetal views. **d** Schematic representation of the drug treatment windows conducted in this study, targeting the period of 4d specification (5 hpf). Number of cases is shown to the right of each bar specifying the time interval. **e** Distribution and percentages of the larval phenotypes observed for each window and experimental condition (WT, wild type). **f** Lateral views of z-stack projections of control and 4–6 hpf SU5402 treated embryos fixed at the larval stage and stained with DAPI (nuclei), phalloidin (F-actin) and Acetylated Tubulin (cilia). Treated larva show a phenocopy of the U0126 radial larval phenotype. On the left, schematic diagrams of lateral views of the controls and radial larva phenotypes. **g** Lateral views of whole mount in situ hybridisation of five cell type markers in control and 4–6 hpf SU5402 treated embryos fixed at the larval stage. Treated embryos develop into radialised larvae with expanded oral ectodermal markers (*gsc*), reduced apical ectodermal markers (*six3/6*), lack of hindgut and trunk mesodermal genes (*cdx* and *twist*, respectively) and normal midgut markers (*GATA4/5/6b*). Insets are ventral views. In **b**, **c**, **f** and **g**, asterisks point to the apical pole. For **d–g**, Supplementary Table 12 reports detailed numbers. Scale bars are 50 µm. Schematic drawings are not to scale. an anus, ao apical organ, ch chaetae, gp gastral plate, mg midgut, mo mouth, pt prototroch.

**Orthology assignment and domain architecture analyses**. Generally, gene orthology was based on the functional annotation of *O. fusiformis* gene repertoire, based on BLAST searches against SwissProt database and Panther searches (functional annotation available at the repository https://github.com/ChemaMD/OweniaGenome). Protein sequences for ERK1/2 and ERK5, WNT ligands, RHOX and related homeoboxes, DLL and JAGGED, FOXH and FOXF genes and FGFR and VEGFR were retrieved by mining publicly available transcriptomes and databases. Multiple protein alignments (available in Supplementary Data 1) were constructed with MAFFT v.7 in automatic mode[63] and poorly aligned regions were removed with gBlocks version 0.91b[64]. Maximum likelihood trees were constructed with either RAxML v.8.2.11[65] or FastTree[66] and visualised with FigTree (https://github.com/rambaut/figtree/). InterProScan 5[67] was used to perform the protein domain composition of DLL.

**Gene isolation and gene expression analyses**. Candidate genes (Supplementary Table 6) and gene markers used to characterise the drug phenotypes were amplified by two rounds of nested PCR using gene-specific primers and a T7 universal primer on cDNA from mixed developmental stages as initial template. Riboprobes were synthesized with the T7 enzyme (Ambion's MEGAscript kit, #AM1334) and stored in hybridisation buffer at a concentration of 50 ng/µl at −20 °C. Colorimetric and fluorescent whole mount in situ hybridisation were performed following an established protocol[3,30]. After probe washes, samples for fluorescent in situ hybridisation were incubated with anti-DIG-POD Fab fragments (ROCHE, #11633716001) and a mouse anti-acetylated α-tubulin antibody (clone 6-11B-1, Millipore-446 Sigma, #MABT868, 1:500) to co-stain cell boundaries. After signal development with a Tyramide Signal Amplification kit (Akoya Biosciences, #NEL742001KT), these samples were washed several times in PTx and treated for secondary antibody

incubation as described above. To characterise in silico gene expression dynamics during development, we used an available stage-specific RNA-seq dataset[33], representing the average expression of the two replicates with the package pheatmap v.1.0.12 available in R, were colour intensity shows the z-score value for each gene.

**Phenotype characterisation and classification**. We used a combination of six morphological and six molecular markers to characterise the phenotypes obtained after all drug treatments (Supplementary Table 3). Briefly, visual inspection and z-stack projections of confocal scanning images were used to assess the presence of chaetae, foregut and hindgut, muscles (in the foregut, chaetoblasts and levators), apical organ (with apical tuft of cilia) and apical organ neurons. To support the presence/absence of these larval structures, we assessed the expression of cell-type-specific molecular markers through colorimetric whole mount in situ hybridisation, namely *six3/6* (apical ectoderm and oesophagus), *gsc* (anterior oral ectoderm), *cdx* (hindgut), *GATA4/5/6* (midgut), *twist* (mesoderm) and *synaptotagmin-1* (neurons). We define the compressed phenotype as a larva with a morphology and molecular patterning like a wild type but with a flattened apical-ventral axis. The radial phenotype lacks hindgut and mesodermal structures and markers, and exhibit radialised expression of the oral ectodermal marker *gsc*, with a reduced number of apical organ neurons and apical organ. The "stubby" phenotype shows bilateral symmetry and all morphological landmarks of a control larva, but the posterior region and mesoderm is underdeveloped. Supplementary Fig. 3d shows schematic representations of control, compressed and radial phenotypes and their respective morphological/molecular landmarks.

**Imaging**. Representative embryos and larvae from colorimetric analyses were imaged with a Leica DMRA2 upright epifluorescent microscope equipped with an

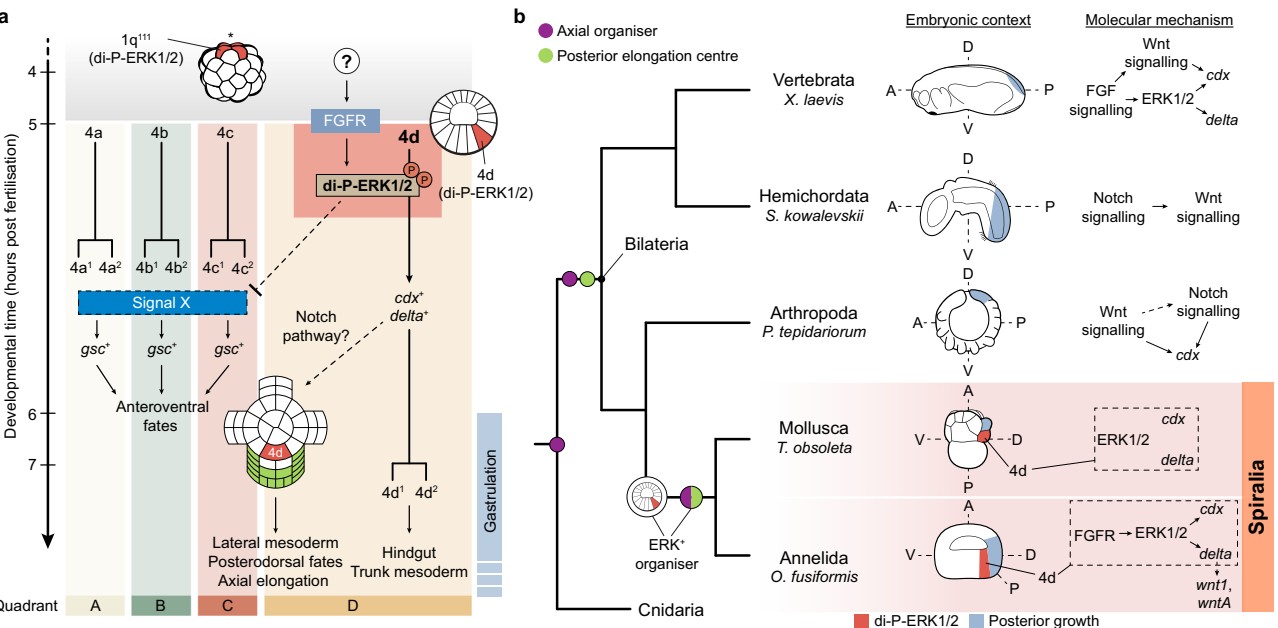

**Fig. 9 ERK1/2 activity in the D-quadrant embryonic organiser is an ancestral spiralian trait. a** Schematic model of FGFR/ERK1/2 activity in the 4d micromere during late spiral cleavage in *O. fusiformis*. At 4 h post fertilisation (hpf), di-phosphorylated ERK1/2 (di-P-ERK1/2) is enriched in the apical most micromeres (1q[111]). At 5 hpf, an unidentified signal activates FGF receptor which di-phosphorylates ERK1/2 in the 4d micromere. This activates *cdx* and *delta* in 4d (becoming the endomesodermal precursor), which also correlates with 4d arresting its cell cycle progression, as 4a–4c cleave. ERK1/2 activation in 4d is required to limit anterior genes (e.g., *gsc*) to the A–C quadrants, yet the exact nature of this interaction is unclear (dotted lines). In addition, ERK1/2 activity in 4d induces the expression of a battery of genes in the D-quadrant and part of the C-quadrant that probably induces lateral mesodermal and posterodorsal fates, ultimately controlling axial elongation. Beginning at this stage, the Notch-Delta pathway is required for normal posterodorsal development and might contribute to transmit the inductive activity of 4d cell to neighbouring cells (dotted line). **b** Schematic drawings of embryos at the time of anteroposterior axial elongation in different bilaterian lineages, with the interactions between FGFR, ERK1/2 and some of the candidate genes identified in this study in each lineage depicted to the left. The similar involvement of ERK1/2 activation in the specification of the D-quadrant embryonic organiser in *O. fusiformis* and other molluscan embryos suggests that the presence of an ERK1/2 positive embryonic organiser is a shared ancestral trait to spiral cleavage. More broadly, ERK1/2 activity induces the expression of a set of genes and signalling pathways that are repeatedly involved in axial patterning and posterior trunk growth in bilaterian embryos. While axial organisation and the formation of a posterior elongation centre are most often spatially and temporally uncoupled in most bilaterian lineages, they become integrated into a single developmental process of embryonic organising activity (i.e., the specification of the D-quadrant embryonic organiser) in spiralians. Drawings are not to scale. A anterior, P posterior, D dorsal, V ventral.

Infinity5 camera (Lumenera), using bright field and differential interference contrast optics. Fluorescently stained samples were scanned with a Leica SP5 LSCM. Confocal stacks were analysed with Fiji and brightness/contrast and colour balance were adjusted in Pixelmator Pro (v. 2.0.3) or Photoshop (Adobe), applying changes always to the whole image and not parts of it. Final figures were designed using Illustrator (Adobe).

**Transcription factor motif identification**. To identify transcription factor motifs in chromatin accessible regions around the *cdx* and *delta* loci, we use two already available replicated Assay for Transposase-Accessible Chromatin using sequencing (ATAC-seq) libraries[33] generated from >50,000 cells from 850–900 coeloblastula at 5 hpf. Briefly, cells were dissociated and lysed in ice cold ATAC lysis buffer (10 mM Tris-HCl pH 7.5, 10 mM NaCl, 3 mM MgCl$_2$, 0.1% (v/v) IGEPAL, 0.1% (v/v) Tween-20, 0.01% (v/v) Digitonin) with the gentle use of a pestle and incubated on ice for 3 min. After a quick wash with ice cold PBS and further resuspension in ice cold ATAC lysis buffer, the lysis reaction was stopped with ice cold ATAC wash buffer (10 mM Tris-HCl pH 7.5, 10 mM NaCl, 3 mM MgCl2, 0.1% (v/v) Tween-20) and mixed by gently inverting the tube three times. The nuclei were then pelleted by centrifugation at 500 g for 10 min at 4 °C in a fixed-angle centrifuge and chromatin tagmentation and library preparation was performed following the Omni-ATAC protocol[68]. Read mapping and peak calling was performed as described elsewhere[33], and transcription factor motif enrichment and footprinting at chromatin accessible regions was performed with HOMER v.4.11[69] and TOBIAS v.0.12.0[70], respectively, using tracks of reproducible peaks between replicates ($p < 0.05$) available at Gene Expression Omnibus with accession number GSE184126 and a public repository (https://github.com/ChemaMD/OweniaGenome). Genomic tracks were plotted with pyGenomeTracks v.2.1[71].

**Reporting summary**. Further information on research design is available in the Nature Research Reporting Summary linked to this article.

## Data availability

The authors declare that the data supporting the findings of this study are available within the paper and its supplementary information files. All raw RNA-seq sequencing data generated in this study is available in the European Nucleotide Archive (ENA) under accession number PRJEB47195. Additionally, this study used the genome assembly and annotation of *Owenia fusiformis*, available at ENA under accession GCA_903813345 and ATAC-seq peaks available at Gene Expression Omnibus with accession number GSE184126 and a public repository (https://github.com/ChemaMD/OweniaGenome).

## Code availability

No custom code was used in this study.

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

## Acknowledgements

We thank Ferdinand Marlétaz, Daria Gavriouchkina, Bruno Vellutini and all members of the Martín-Durán lab for support and valuable comments on the manuscript, as well as Sandra Álvarez-Carretero for help with RNA-seq analyses, the staff at Station Biologique de Roscoff for their help with collections and animal supplies, and the Oxford Genomics Centre at the Wellcome Centre for Human Genetics (funded by Wellcome Trust grant reference 203141/Z/16/Z) for the generation and initial processing of RNA-seq sequencing data. This work was funded by a European Research Council Starting Grant (action number 801669) to J.M.M.-D. and a Queen Mary, University of London studentship to O.S.

## Author contributions

O.S., A.M.C.-B. and J.M.M.-D. conceived and designed the study. O.S., A.M.C.-B., Y.L. and J.M.M.-D. performed all experimental approaches and critically analysed the data. O.S., A.M.C.-B. and J.M.M.-D. wrote the manuscript. All authors read and commented on the manuscript.

## Competing interests

The authors declare no competing interests.
