## [Peer Review File · Nature Communications]

ERK1/2 is an ancestral organising signal in spiral cleavageReviewers' Comments:

Reviewer #1:

Remarks to the Author:

SUMMARY

Spiralians are a large group of organisms that show stereotypic development, similar to *C. elegans* or tunicates like *Ciona intestinalis*. This allows comparisons of specific embryonic cells and their roles in establishing body plans across many Spiralian species. This makes Spiralians a powerful system and exciting opportunity for studying evolution of development. The work reported here by Seudre and colleagues takes advantage of a basal annelid model (*Owenia fusiformis*) and asks if this species has ERK activity that acts as an embryonic organizer, which is present in other Spiralians (such as mollusks) but not in many other annelids that have been studied to date. Through this comparison across these two Spiralian clades (mollusks vs annelids) they conclude that ERK signaling as an embryonic organizer is an ancestral feature in Spiralia. I think this is a very interesting study, with beautiful and clear experiments that mechanistically test function, and show that: 1) ERK (and FGR upstream to ERK) indeed is active in the organizer cell 4d, and it regulates patterning. 2) By inhibiting ERK (using small molecules) the axial patterning is affected as indicated by radialized embryos that also lost specification of postero-dorsal mesodermal and ectodermal lineages, and downstream gene expression patterns are disrupted. 3) They show that the radial phenotype and loss of expression patterns are a result of failing to specify the organizer cell (4d). Therefore, data presented support the results, the methods are sound, lots of supplementary information is provided.

Where I think the manuscript falls short is the discussion. Authors don't give us enough general conceptual information and explanation on what this conserved pathway in a basal annelid means in the big scheme of embryogenesis and body plan evolution. The discussion starts with very detailed, jargon-rich text, focused on the minute details of spiralian embryogenesis, which will be exciting for anyone interested in spiralian embryology, but to make the findings more exciting to a broader Nature Communications reader, discussion needs to zoom out and place all these findings in a broader context. It is very interesting that a developmentally-early signaling pathway is conserved in a basal annelid and snails (mollusks), while you still end up making a wormy body, not a snail! I am surprised there is not any discussion on this. I am left with wanting help with placing all this beautiful data in a larger context of animal evolution of development. Can the authors tell the reader (who is most likely not familiar with annelids or spiralians because this is Nature Communications) more clearly why their findings are so exciting and important? Basically, the substance is there, experiments are robust and beautiful, but the larger context is missing (which is possible to remedy by re-writing some sections in the manuscript, especially discussion).

MAJOR COMMENTS

- 1) I strongly suggest discussing your findings in a broader context and not starting your discussion's first paragraph with so much specific spiralian/annelid embryology jargon (which I personally find interesting, but I think you will benefit from imagining what would a reader who is not familiar with annelids/spiralians would like to see as the first few sentences into discussion).
- 2) I want to invite the authors to reconsider the use of terms (I assume they coined) "conditional[ly] cleaving" vs "autonomous[ly] cleaving". The embryos aren't conditionally cleaving. They will cleave, that is what they do. The specification of fates is autonomous or conditional. I think in their current form, the terms are confusing. Maybe it would be better to simply say "conditional annelids" vs "autonomous annelids", if the concern is to have a concise way of referring to the conditional vs autonomous developmental modes.
- 3) Line 84 and after (1st section in the Results): It would help to explicitly explain the aim of this section. Basically, you need to show that there is a blastomere in *Owenia* embryos that is the potential organizer, this cell is 4d, and that it has ERK expression. Help the reader (who is not familiar with

spiralian/annelid development) beforehand by telling them explicitly.

4) Lines 94-99 (starting with "this cell is one of the fourth quartet..."): All of this info could become supplementary. I would keep this part more jargon-free and general. Basically, because this is an equal cleaving embryo, you cannot label the blastomeres as ABCD. But you need to determine the D lineage, which makes the organizer in other spiralian. You find the cell with the unique behavior (start of bilateral symmetry), which safely allows you to call it 4d, and makes it a good candidate for being the organizer. And then you show that that very same cell also expresses the active form of ERK (di-P-ERK). Therefore, I would make Fig 2A much larger, and put most of Fig 2B into supplementary, along with the text in lines 94-99, and keep it more general in the main text, for accessibility. I would keep the schematics in 1C in the main figure (they are helpful). This will also help making Fig 2 more readable, the labels are painfully small, impossible to read in some instances.

5) ATAC seq protocol details should be provided. It appears that there are no details about how the authors dissociated cells and isolated nuclei for ATACseq in *Owenia*. (starting at Line 437)

6) I thought it was somewhat unclear what criteria the authors used for categorizing the phenotypes as compressed vs radial vs wild type? For example, in Fig S2, to the untrained eye, these are hard to tell. It would be helpful to add these criteria (maybe with some drawings) into this supplementary figure, and explain better how you categorized samples (probably in the methods, or if space is an issue, in the supplementary information). This is such an important part of the data, and we need to know how the authors made the decision for the samples in categorizing them. Were they systematic about this scoring/categorizing, or was it just "measuring by eye"?

MINOR COMMENTS

1) The supplementary file names were not properly labeled and it was confusing and hard to find which table was which.

2) Figure 2A: Labels in this figure are painfully too small. Also, it would help to have individual labels for individual images and refer to them specifically in the text.

3) I found Table S1 to be very helpful, thank you.

4) In Figure 3C, neither in the figure itself nor in the legend, the time of fixation (how old the samples we are looking at) is stated. Please include this information.

5) In Figure 3D, the ectodermal expression for six3/6 is claimed to be reduced, but I am having a really hard time seeing this. I am not convinced, so I suggest providing more images, or a different view, or different zoom/labeling.

6) Also in Figure 3D, in the legend, authors mention a midgut gene (*GATA4/5/6b*), but I do not see this in the figure. If it is in supplementary, please indicate.

7) Line 131: I am confused, why are the authors citing REF 29 here? Was there an experiment in that paper that has this same treatment?

8) Lines 251-252: This is a confusing sentence, I suggest revising it.

9) Line 271: "...compared to the human *FGFR1* ortholog (Figure 7B)." It seems like maybe this panel in Fig. 7B was moved to Fig S5C?

10) Line 301: Please provide citations for these "traditional views".

11) Line 339: So, the enrichment of ERK is not unique to *Owenia*, but the activity is. It would be nice (if possible) to have another column in Fig 1C, between Specification Mode and ERK1/2 axial organizer columns, ERK expression in the organizer cell. This way, we can see if there is expression (like in *Hydroides*) but not activity/function as an organizer.

12) Figure S1C: DAPI insets in this panel are very hard to see. I suggest changing the color to cyan or something that has more contrast against the black background.

13) In Figure 4D, there are colored spots under the Condition column. It is unclear what these are, I do not see a reference to a supplementary figure for these either. Quite confusing.

14) In several places in figures (ex: Fig 4B, Fig 5E) the authors label axes as "No differentially expressed genes" where they mean "Number of" I think. It is confusing. I suggest either adding "of" after "No", or maybe using a hash symbol?

15) Figure 5E: This is confusing and I am not sure what I am looking at. Very little help in the legend about how to read this panel.

Reviewer #2:

Remarks to the Author:

This manuscript is an investigation of the embryonic organizer in the annelid *Owenia fusiformis*. The cellular identity and molecular character of embryonic organizers have been investigated in related animals, specifically molluscs and annelids that exhibit a similar spiral cleavage program to *O. fusiformis*. However, this is the first study to investigate the embryonic organizer in an equal cleaving annelid. This study characterizes the pattern of cells that show a phosphorylated form of ERK1/2, an intracellular signal transducer that becomes active upon phosphorylation. One of the cells that shows a phosphorylated form of ERK1/2 is 4d, a cell that has organizing activity in other animals. The core of this study includes a set of chemical inhibition experiments to identify the timing of D quadrant specification as well as the signal transduction pathway and molecular signals utilized in D quadrant specification and organizing activity. Most of the interpretations are based on the drug inhibition data. Perturbation of the ERK1/2 pathway uses a commercially available reagent (UO126) at multiple time intervals during early embryogenesis to perturb signal transduction and implicate this signal transduction cascade in organizing activity. Phenotypes are analyzed morphologically and by in situ hybridization using multiple molecular markers. The authors interpret the phenotype as radial and a consequence of misspecification of 4d (and of the D quadrant). This is followed by transcriptome profiling of embryos treated with UO126 or BFA and ATAC-seq to identify downstream targets of ERK1/2 signal transduction. Differential expression analysis is validated by in situ hybridization. The authors argue they have identified the receptor that specifies the D quadrant by performing chemical inhibition studies of FGFR signaling (commercially available SU5402), and demonstrating that the resulting phenotype is the same as when embryos are exposed to the ERK1/2 inhibitor. This study also includes report of a genome sequence for *O. fusiformis*, RNA-seq data for 12 developmental time points between the oocyte and the larval stage, differential gene expression analysis, ATAC-seq and developmental characterization of numerous genes at multiple developmental stages. From these data, the authors interpret *O. fusiformis* to represent the ancestral condition of spiral cleaving animals from the rationale that ERK1/2 signaling is implicated in organizer signaling in molluscs and the annelid *O. fusiformis*. The authors also state that their findings indicate that both autonomous and conditional mechanisms specify the D quadrant in *O. fusiformis*, a finding that sets it apart from other equal cleaving forms.

This study investigates an important topic through the lens of developmental biology and directly relates to the evolution of animal diversity. The spiralian represent nearly one third of animal diversity and therefore, *O. fusiformis* is an excellent choice of animal study system and its study is likely to reveal insights into evolution of evolution of animal body plans.

This manuscript clearly includes an impressive amount of data, some of which are more relevant to the study than others. Data for several aspects of this study are convincing. For example, chemical inhibition studies are performed with appropriate controls, range of concentrations, sample sizes, etc. The quality of in situ hybridization images is high and phenotypic interpretations are drawn from the use of numerous markers. Data included but not completely essential to this study will be useful to the community (e. g. genome sequence, RNA Seq, transcription of ERK1/2 gene, etc), but makes it challenging for the reader to follow the important thread of this study. The writing is confusing in parts, particularly the evolutionary arguments in the discussion. Moreover, to accommodate all the data, the figures are complex, and some are difficult to follow or comprehend. Many figure legends do not contain sufficient detail (see detailed comments below). Together, these issues make it challenging to evaluate all of the data presented.

Yet, there are important pieces of data missing from this study. For example, given the large number of in situ expression patterns presented, it is surprising to not see the expression of FGFR since this is a targets of the drug SU5402. Is it in the same cell as the cell that shows activation of ERK1/2? This is an important piece of information to corroborate their SU5402 drug inhibition data.

The reviewer is not convinced that the authors can really identify 4d as the cell with organizer activity in *O. fusiformis*. The typical demonstration of such a function would be to perform single cell transplantation or deletion manipulations. This study does not include any such manipulations, and therefore, the authors need to update the language in the manuscript to acknowledge that their interpretations are based upon indirect deduction and that 4d 'may be' the cell with organizing activity but that future direct experimental manipulations are required to confirm.

The authors observe the same phenotype using all three drugs that have very different target pathways/molecules. How do the authors know that the resulting phenotype does not reflect toxicity or some other non-specific effect? Are there other chemical inhibition studies for this animal that give a distinct phenotype and could be cited? If not, demonstration of specificity is an important control to include.

Although the results presented are interesting, the proposed evolutionary argument is not strong. First, the authors argue that organizing activity in the annelid *O. fusiformis* is homologous to that of molluscs, but not to the annelids *Platynereis dumerilli*, *Tubifex tubifex*, *Chaetopterus pergamentaceus*, and *Capitella teleta*. Their results that implicate ERK1/2 signal transduction provides support for this argument. What is not clearly acknowledged is that the pattern of ERK1/2 activation varies across mollusc species and none of the published mollusc patterns are the same as the pattern observed in *O. fusiformis*. To my knowledge, the only published pattern of ERK1/2 activation that is the same as that of *Owenia* is found in another equal cleaving annelid, *Hydroides hexagonus*, an annelid within the *Sedentaria* clade. This calls into question the authors proposed evolutionary scenario that annelids lost ERK1/2 activation as part of D quadrant activation (Fig 6B). Second, a stronger evolutionary argument would include multiple pieces of evidence. For example, the cellular mechanism of D quadrant specification is known for some molluscs (e. g. *Patella* and *Lymnea*) and involves critical cell contacts between the macromeres and first quartet micromeres. At a minimum, this literature should be discussed.

There are numerous additional issues that need to be addressed and are outlined below.

Detailed comments:

- The authors should acknowledge that unequal cleavage may be ancestral for annelids. Even if the authors disagree, the published alternate opinion should be cited (Dohle 1999).
- Line 29, the wording 'instructing role of ERK1/2...' is misleading. Activation of ERK1/2 occurs when a cell receives a signal and is NOT a direct sign of a signal being given or 'instructing'
- Line 38, 'instruct neighbouring tissues' should be 'instruct neighbouring cells'. Tissues have not yet formed at this stage.
- Line 83 - 85, overstatement to say that ERK1/2 activity spread convergently to some annelid lineages. Best existing candidate of a convergent mechanism is *Hydroides* and to my knowledge, there is no functional data for *Hydroides* (if there is, please cite and add statement) a requirement to make a statement about function(e.g. 'activity').
- Line 92, citation 32, is this the correct citation since the title suggests that the study is about two other annelid species. Is it really about *O. fusiformis*?
- Line 101, why show transcript expression for ERK1/2 ortholog when the relevant information is whether or not ERK1/2 is phosphorylated?
- The authors identify a single cell that is di-P-ERK1/2 positive (Fig 1D, Fig 2B). How do the authors really know it is 4d if it is an equal cleaving form? The size of 4qs appear to be the same (Fig 1E).
- Line 121, are the two daughter cells of 4d di-P-ERK1/2 positive? Or is signal lost once 4d divides? How long does the di-P-ERK1/2 form persist in 4d or its daughters?
- Lines 124 - 126, Cannot tell from wording what exactly 'cellular dynamics' refers to.
- Lines 149 - 152, exposure to UO126 is from 0.5 - 5hr, which encompasses 3 phases of di-P-ERK1/2 positive cells: animal pole cells, then '4d' then 4d plus other vegetal 2q cells. Therefore, wording of this statement is an oversimplification since the resulting phenotype is likely a reflection of interfering

with signaling at all three phases. Please expand statement to acknowledge that e. g. the apical tuft phenotype may be a consequence of inhibiting activation of P-ERK1/2 in the 1q111 cells and other defects may reflect inhibition of 2q12 and 2q22 cells, perhaps some of the posterior structures.

-Line 160, what does a 'compressed morphology' mean? Compressed along what axis? This is not a standard term and therefore it needs to be defined.

-Line 169, 'ERK1/2 activity is required for normal embryonic activity'... this does not fit very well with the pattern of activation as shown by immunohistochemistry. How do the authors explain this?

-Line 202, the following developmental genes are not to my knowledge involved in axial patterning in spiralian: *six3/6*, *gsc*, *cdx*, *AP2*, *foxQ2*. Need to add rationale, supporting evidence or modify language to state that these genes are markers of particular cell and tissue types.

-Line 202, 'recurrently' is not a word.

-Line 241, this statement does not obviously follow from the previous information in the paragraph 'ERK1/2 di-phosphorylation in 4d seems to delay cell cycle progression in this cell

with respect to 4a-c'. How does ATAC sequence data relate to cell cycle progression?

-Line 249 - 250, how is the fate of 3q cells in an echinuran relevant to this study?

-Line 262, 4d does not appear to be larger than the other 4q cells (Fig 1E) as stated in the text. A clear image showing this needs to be provided to make such a claim.

-Line 339, add '6hr' to figure reference 'Figure 1E'. It is challenging to find information the figures referenced in the text since unique identifiers of individual panels are lacking. This is a general challenge throughout the figures.

-Line 354, '...indicating that unlike most annelids' is an overstatement. The manuscript cited is that of a single species. Although there may be data for 1 - 2 other species, it is prudent for the authors to acknowledge that there are almost 20,000 described annelids. Change 'most' to 'some'.

-Line 486, Add a reference to support the agar plug DNA extraction method.

-Line 980, '.BFA and UO126 treated larva' Should be 'larvae'. This is incorrect as worded since the larvae themselves were not exposed to the drug. The reviewer's understanding is that embryos were treated with drug and then raised to larval stages for phenotypic analysis.

Figures:

There is a lack of a unique identifier (e. g. panel letter or label) for each image/graph/schematic presented. As a reviewer, it is difficult to refer to individual parts of figures since one letter may refer to up to 14 different images.

Figure 1B, recommend including the annelids specifically mentioned in the text in the phylogenetic tree to give the reader a better appreciation of their phylogenetic distribution and interpretations that are drawn. For example, representatives of both Sedentaria and Errantia are cited, as well as the early branching species *Chaetopterus* (reviewed in Weigert and Bleidorn 2016).

Figure 2, what are the inserts in panels shown in B? in D? please add info to legend. Same question for Figure S2C. Check all figure legends.

Fig 2C, 2D, recommend adding ventral view insets for the control images to compare with the ventral views of the treated animals.

Fig 2D Chaetae should be labeled since they are mentioned as one of the scored characters in the text.

Fig. 2E, right, observed larval phenotype graphic is very difficult to understand! There is almost no text in the legend that explain this graphic and how it relates to the left part of the panel.

Fig. 2E, far fewer cases (n=) for critical time interval (4 - 6 hr) than for other time intervals. Why? Also, do these few cases represent independent replicates? Please add to legend that number of cases is shown to the right of each bar specifying the time interval.

Fig 3B, Y axis label is confusing 'no differentially expressed genes' should be '# differentially expressed genes' or 'No. differentially expressed genes'

Fig 3B, what do the circles, grey circles, and lines grey connecting lines at the bottom of the figure panel denote? Add heading and/or info to legend. What are the exact pairwise comparisons that result in up and down regulated genes shown for each of the bars in the bar graph?

Fig 3C, how do the authors explain the substantial differences in identity of differentially expressed genes when embryos are treated with BFA versus UO126 with the same resulting phenotype? This

needs to be addressed.

Fig 3D, what is the coding of the colored dots on the left side of the panel? What does inclusion of a dot indicate?

Fig 3D, Should be 'vertical' dotted line, not horizontal dotted line as indicated in legend. Perhaps this refers to a previous version?

Fig 4, hard to see cell boundaries (AcTub) in fluorescent panels in A, B and C. Better in some panels in F.

Fig 4D, are *cdx* and *delta* the only developmental genes identified with open chromatin in this ATAC-seq analysis? If so, please state in text. If not, why were these genes highlighted and not others?

Fig 4E is difficult to understand and there is not sufficient explanation in the legend. Why is the heading for the schematic 'no cellular contacts'? It appears that the point of the schematic is cellular contacts?

Fig 5C, right side of panel is very hard to decipher. Needs explanatory text. Same issue as in Fig. 2E

Fig 6 legend Lines 936 – 7, activation in 4d by ERK1/2 limits an anteriorizing signal in the A – C quadrants. Rather than signaling from 4d, is it possible that the expansion of *gsc* expression is a more direct affect of the interference of di-P-ERK1/2 in the 1q111 cells?

Fig S1C, is this really low level activation in early cleavage stages or lack of specific signal? What is needed are negative controls such as a no primary antibody and no antibody to see signs of any endogenous reaction or background.

-Fig S2C, BFA treatment, looks like see expansion of neural tissue? How do the authors interpret this phenotype? Is this related to the di-P-ERK1/2 in the 1q111 cells?

Fig S4, what is the rationale for organizing the gene expression patterning in the groupings shown? Need to add reference to individual panel letters to provide this information.

Fig S5B, What views of the embryos are shown in the panels?

Fig S5B, in legend it is stated that *foxH* is expressed in cells giving rise to mesodermal derivatives. Does this mean that these are daughters of 4d, possibly ML and MR, which generate most mesoderm in spiralian. If so, is this not a contradiction with a statement that 4d does not divide during this time interval?

Fig S6C, should be 'vertical' dotted line, not 'horizontal' dotted line.

Fig S6F, legend says that there is reduction of *six3/6* expression but the images actually show an expansion in the area of expression. Please correct/explain.

Reviewer #3:

Remarks to the Author:

This is a sound study that will be of interest to specialists within the field of evolutionary developmental (evo-devo) biology. The work provides evidence that FGFR and ERK1/2 signaling specify and control the activity of the embryonic organizer in the spiral-cleaving annelid *O. fusiformis*, which undergoes a conditional mode of embryonic development. The paper is beautifully written. The authors conduct precise gene expression analyses at early stages of development to identify where and when phosphorylated Erk is present, followed by pharmacological inhibition of Erk activity to assess gene expression and morphological effects. The RNA-Seq data are independently validated with RNA ISH. Lastly, pharmacological inhibition of FGFR phenocopies the effects observed upon drug inhibition of Erk. Enough detail is provided in methods for the work to be reproduced by others. Most of the data support the conclusions of this study. However, additional evidence is needed to strengthen some of the key claims, as outlined below:

1. Figure 2 describes a detailed expression analysis for di-phosphorylated ERK1/2. Because only cell positional criteria were used to infer the identity of the 4d cell, additional molecular markers for 4d could be used to strengthen the point that di-phospho ERK1/2 is enriched in 4d.

2. The analysis of the two drugs, BFA and U0126, is detailed and well-presented. However, drugs can have numerous off-target effects. Is there a way in *O. fusiformis* to disturb the phosphorylation of ERK1/2 by either using a masking antibody against the phosphorylation site (s), or RNAi (shRNA) to

downregulate MEK?

3. Lines 216-222: The conclusions drawn are incorrect for two reasons. First, ATAC-Seq provides an assessment of open the chromatin at specific regions. This method only allows inferences to be made about which transcription factor motifs are present in a specific region of DNA that is "open". ATAC-Seq cannot identify transcription factors bound to the accessible regions. ChIP-Seq can do this. Second, to test causality and the author's claim in line 221 (ERK phosphorylation in 4D seems to control the activity of transcriptional regulators) the ATAC-Seq experiments needs to be conducted in the presence of U0126, or other more specific phospho-ERK manipulations proposed in point #2. According to the authors hypothesis, an expected outcome would be that the open chromatin regions of cdx and delta will be "closed" upon U0126 treatment.

4. A central pathway in this paper appears to be the Notch pathway, as delta is downstream of FGFR/ERK. Pharmacological inhibitors for Notch are available and readily used in evo-devo projects. They authors are encouraged to used them and test whether Notch inhibition partially phenocopies the effects observed upon FGFR/ERK inhibition.

5. The claims related to BFA treatment should be toned done throughout the paper, as BFA is a global protein trafficking inhibitor that affects many many proteins. For example, in line 251-252 it is stated that gsc expression requires inductive signals based on the BFA data. However, the trafficking of proteins made from maternally deposited mRNAs will also be affected by BFA, hence the requirement of autonomous signals cannot be ruled out for gsc expression.

6. The data with the FGF inhibitor make the compelling argument that FGFR is upstream of phospho-ERK. However, these data could be presented in more detail. For example, it would help the reader if the phenocopy of the SU5042 and U0126 treatments is discussed in more detail - morphological and gene expression effects can be compared and contrasted. Again, the BFA treatment data should be toned down here due to its broad effects on protein trafficking.

Response to reviewers

We would like to thank the three reviewers for their positive appraisal of our manuscript, and for providing a range of thoughtful comments and suggestions that we believe have significantly improved our work. Following the point raised by reviewer #3, we now include a functional characterisation of the role of the Notch-Delta pathway before and after the specification of the D-quadrant organiser, which demonstrates a role of this signalling pathway in posterodorsal development and thereby supports our originally proposed model. Moreover, we have expanded our analysis of ERK1/2 activity to cover the gastrulation phase (from 6 to 9 hours post fertilisation), which allows us to monitor the behaviour of the MR/ML cells (4d daughters) and their relationship with the expression of the *foxH* gene (reviewer #2 concerns). We also provide a time course of FGFR expression to support the functional phenotypes observed after its inhibition and negative controls for the immunohistochemistry against di-phosphorylated-ERK1/2 at early cleavage stages (reviewer #2 comments). In addition to these new datasets, we have thoroughly changed the Discussion (reviewer #1 and #2 suggestion) to make it broader and qualify our evolutionary comparisons, as well as simplified and given more context to the first section of the Results (reviewer #1 comment). We have also expanded on the comparisons between the different drug phenotypes (reviewer #3 suggestion), providing clearer classification criteria (reviewer #1 comment) and improved all figure legends (reviewer #1 and #2 points). Finally, we incorporated all other text and figure suggestions, reformatted all supplementary tables for easier access, adapted the manuscript to the journal format requirements, and distributed all main findings into nine display items and seven supplementary figures (three more than in the original submission), which has allowed us to increase panel and font sizes and make figures simpler and more focused (reviewer #2 suggestion).

Below we provide a detailed point-by-point response to each of the reviewers' concerns.

Reviewer #1 (Remarks to the Author):

Spiralians are a large group of organisms that show stereotypic development, similar to C. elegans or tunicates like Ciona intestinalis. This allows comparisons of specific embryonic cells and their roles in establishing body plans across many Spiralian species. This makes Spiralians a powerful system and exciting opportunity for studying evolution of development. The work reported here by Seudre and colleagues takes advantage of a basal annelid model (Owenia fusiformis) and asks if this species has ERK activity that acts as an embryonic organizer, which is present in other Spiralians (such as mollusks) but not in many other annelids that have been studied to date.

Through this comparison across these two Spiralian clades (mollusks vs annelids) they conclude that ERK signaling as an embryonic organizer is an ancestral feature in Spiralia. I think this is a very interesting study, with beautiful and clear experiments that mechanistically test function, and show that: 1) ERK (and FGR upstream to ERK) indeed is active in the organizer cell 4d, and it regulates patterning. 2) By inhibiting ERK (using small molecules) the axial patterning is affected as indicated by radialized embryos that also lost specification of postero-dorsal mesodermal and ectodermal lineages, and downstream gene expression patterns are disrupted. 3) They show that the radial phenotype and loss of expression patterns are a result of failing to specify the organizer cell (4d). Therefore, data presented support the results, the methods are sound, lots of supplementary information is provided.

Where I think the manuscript falls short is the discussion. Authors don't give us enough general conceptual information and explanation on what this conserved pathway in a basal annelid means in the big scheme of embryogenesis and body plan evolution. The discussion starts with very detailed, jargon-rich text, focused on the minute details of spiralian embryogenesis, which will be exciting for anyone interested in spiralian embryology, but to make the findings more exciting to a broader Nature Communications reader, discussion needs to zoom out and place all these findings in a broader context. It is very interesting that a developmentally-early signaling pathway is conserved in a basal annelid and snails (mollusks), while you still end up making a wormy body, not a snail! I am surprised there is not any discussion on this. I am left with wanting help with placing all this beautiful data in a larger context of animal evolution of development. Can the authors tell the reader (who is most likely not familiar with annelids or spiralian because this is Nature Communications) more clearly why their findings are so exciting and important? Basically, the substance is there, experiments are robust and beautiful, but the larger context is missing (which is possible to remedy by re-writing some sections in the manuscript, especially discussion).

ACTION: We thank the reviewer for the positive appraisal and as we indicate in the point below, we have extensively modified the Discussion section to give broader context to our findings, while still addressing the concerns raised by the other reviewers regarding evolutionary interpretations (see below).

MAJOR COMMENTS

1) I strongly suggest discussing your findings in a broader context and not starting your discussion's first paragraph with so much specific spiralian/annelid embryology jargon (which I personally find interesting, but I think you will benefit from imagining what would a reader who is not familiar with annelids/spiralian would like to see as the first few sentences into discussion).

ACTION: As indicated above, we have thoroughly changed the Discussion, and removed the first paragraph in the original submission, which the reviewer found too niche. We now start this section with two general statements on the significance of our work. The rest of the first paragraph briefly highlights the key developmental features of our study on *Owenia* (visually represented in Fig. 9a), before we address the evolutionary implications of our work in the context of Spiralia (second paragraph) and Bilateria generally (third paragraph). In addition, to back our developmental comparisons with other bilaterian lineages (schematically depicted in Fig. 9b), we now include a literature review of the role of FGFR/ERK/Notch in axial patterning in a range of non-Spiralia bilaterian lineages in Supplementary Table 14.

2) I want to invite the authors to reconsider the use of terms (I assume they coined) "conditional[ly] cleaving" vs "autonomous[ly] cleaving". The embryos aren't conditionally cleaving. They will cleave, that is what they do. The specification of fates is autonomous or conditional. I think in their current form, the terms are confusing. Maybe it would be better to simply say "conditional annelids" vs "autonomous annelids", if the concern is to have a concise way of referring to the conditional vs autonomous developmental modes.

ACTION: We agree with the reviewer and have amended the text as suggested.

*3) Line 84 and after (1st section in the Results): It would help to explicitly explain the aim of this section. Basically, you need to show that there is a blastomere in *Owenia* embryos that is the potential organizer, this cell is 4d, and that it has ERK expression. Help the reader (who is not familiar with spiralian/annelid development) beforehand by telling them explicitly.*

ACTION: Thanks for the suggestion; we have thoroughly rewritten the first section of the Results as proposed (lines 87-108), explaining better the purpose of the analyses before describing morphological and immunohistochemical observations in detail.

4) *Lines 94-99 (starting with “this cell is one of the fourth quartet...”): All of this info could become supplementary. I would keep this part more jargon-free and general. Basically, because this is an equal cleaving embryo, you cannot label the blastomeres as ABCD. But you need to determine the D lineage, which makes the organizer in other spiralian. You find the cell with the unique behavior (start of bilateral symmetry), which safely allows you to call it 4d, and makes it a good candidate for being the organizer. And then you show that that very same cell also expresses the active form of ERK (di-P-ERK). Therefore, I would make Fig 2A much larger, and put most of Fig 2B into supplementary, along with the text in lines 94-99, and keep it more general in the main text, for accessibility. I would keep the schematics in 1C in the main figure (they are helpful). This will also help making Fig 2 more readable, the labels are painfully small, impossible to read in some instances.*

ACTION: Thanks for the insightful comment; we modified Figure 2 as suggested, and former panel B is now a stand-alone Supplementary Fig. 1. In addition, and as indicated in the point above, we amended the first section of the Results.

5) *ATAC seq protocol details should be provided. It appears that there are no details about how the authors dissociated cells and isolated nuclei for ATACseq in Owenia. (starting at Line 437)*

ACTION: We provide now more detail on how we prepared ATAC-seq libraries for *Owenia* (lines 557-568). These datasets are part of a large genome regulatory analysis during *O. fusiformis* development that is now publicly available (including all associated datasets) in bioRxiv (<https://doi.org/10.1101/2022.02.05.479245>), Gene Expression Omnibus (accession number GSE184126) and a public repository (<https://github.com/ChemaMD/OweniaGenome>). Given reviewer #2's suggestion to focus the manuscript on the key experimental elements to study the role of ERK1/2, we have streamlined the Materials and Methods section and removed the parts that are not essential datasets to this study and are now explained in more detail in the bioRxiv paper.

6) *I thought it was somewhat unclear what criteria the authors used for categorizing the phenotypes as compressed vs radial vs wild type? For example, in Fig S2, to the untrained eye, these are hard to tell. It would be helpful to add these criteria (maybe with some drawings) into this supplementary figure, and explain better how you categorized samples (probably in the methods, or if space is an issue, in the supplementary information). This is such an important part of the data, and we need to know how the authors made the decision for the samples in categorizing them. Were they systematic about this scoring/categorizing, or was it just “measuring by eye”?*

ACTION: Following the reviewer's suggestion, we now include schematic drawings for all phenotypes in Supplementary Fig. 3, a new section in Materials and Methods to define the criteria used to categorise each phenotype (lines 528-545), as well as a Supplementary Table (Supplementary Table 3) with the criteria and given scores.

MINOR COMMENTS

1) *The supplementary file names were not properly labeled and it was confusing and hard to find which table was which.*

ACTION: Apologies; we have now correctly labelled all supplementary files.

2) *Figure 2A: Labels in this figure are painfully too small. Also, it would help to have individual labels for individual images and refer to them specifically in the text.*

ACTION: We have amended Figure 2 as suggested above, increasing the font size of labels, and considering each individual image as a separate panel to improve their reference in the text (as also suggested by reviewer #2). Former Figure 2B is now a larger, stand-alone figure as Supplementary Fig. 1, which allowed us to increase font size there too to improve readability.

3) *I found Table S1 to be very helpful, thank you.*

RESPONSE: Thanks. We re-formatted Supplementary Figures and Tables as a single PDF that is of easier access than the originally submitted excel sheets.

4) *In Figure 3C, neither in the figure itself nor in the legend, the time of fixation (how old the samples we are looking at) is stated. Please include this information.*

ACTION: We added this information in the figure legend (lines 812-813).

5) *In Figure 3D, the ectodermal expression for six3/6 is claimed to be reduced, but I am having a really hard time seeing this. I am not convinced, so I suggest providing more images, or a different view, or different zoom/labeling.*

ACTION: We have added an inset with a zoom of the expression of *six3/6* in the apical ectoderm of radial larvae. While the endodermal expression of *six3/6* is not affected (and probably even expanded), its ectodermal expression in the apical region is reduced. In the main panels, a dotted line separates the ectodermal from the endodermal expression.

6) *Also in Figure 3D, in the legend, authors mention a midgut gene (GATA4/5/6b), but I do not see this in the figure. If it is in supplementary, please indicate.*

ACTION: This referred to the original Supplementary Fig. 2. We have removed that statement from the legend.

7) *Line 131: I am confused, why are the authors citing REF 29 here? Was there an experiment in that paper that has this same treatment?*

RESPONSE: Former Ref 29 (Carrillo-Baltodano *et al.* 2021; DOI: <https://doi.org/10.1186/s13227-021-00176-z>; now Ref 30) describes the expression of *synaptotagmin-1* during *O. fusiformis* embryogenesis, demonstrating that this gene is a neuronal marker localised in the apical organ. We use this reference here to back the use of this gene as a marker to explore neural development around the apical organ in control versus treated embryos.

8) *Lines 251-252: This is a confusing sentence, I suggest revising it.*

ACTION: We have rephrased this sentence, removing inferences of potential inductive signals based on BFA treatment, as suggested by reviewer #3 (see point below). The sentence now reads as: “The expression of *gsc* expression is independent of ERK1/2 activity, consistent with its location outside the D-quadrant, but is downregulated after BFA treatment (Supplementary Data).” (lines 271-273).

9) *Line 271: “...compared to the human FGFR1 ortholog (Figure 7B).” It seems like maybe this panel in Fig. 7B was moved to Fig S5C?*

10) *Line 301: Please provide citations for these “traditional views”.*

11) *Line 339: So, the enrichment of ERK is not unique to Owenia, but the activity is. It would be nice (if possible) to have another column in Fig 1C, between Specification Mode and ERK1/2 axial organizer columns, ERK expression in the organizer cell. This way, we can see if there is expression (like in Hydroides) but not activity/function as an organizer.*

12) *Figure S1C: DAPI insets in this panel are very hard to see. I suggest changing the color to cyan or something that has more contrast against the black background.*

ACTION: We have amended these four points as suggested (the sentence on “traditional views” was removed when rewriting the Discussion, see point above).

13) In Figure 4D, there are colored spots under the Condition column. It is unclear what these are, I do not see a reference to a supplementary figure for these either. Quite confusing.

ACTION: We amended the figure legend to indicate what the condition column and the dots indicate (lines 843-846).

14) In several places in figures (ex: Fig 4B, Fig 5E) the authors label axes as “No differentially expressed genes” where they mean “Number of” I think. It is confusing. I suggest either adding “of” after “No”, or maybe using a hash symbol?

ACTION: We amended the panel as suggested (No. of ...)

15) Figure 5E: This is confusing, and I am not sure what I am looking at. Very little help in the legend about how to read this panel.

ACTION: We have changed the visualization of this data to a bar plot, which we think better illustrates that 4d cell has more cell-cell contacts than the other 4q micromeres at 5.5 hours post fertilisation.

Reviewer #2 (Remarks to the Author):

*This manuscript is an investigation of the embryonic organizer in the annelid *Owenia fusiformis*. The cellular identity and molecular character of embryonic organizers have been investigated in related animals, specifically molluscs and annelids that exhibit a similar spiral cleavage program to *O. fusiformis*. However, this is the first study to investigate the embryonic organizer in an equal cleaving annelid. This study characterizes the pattern of cells that show a phosphorylated form of ERK1/2, an intracellular signal transducer that becomes active upon phosphorylation. One of the cells that shows a phosphorylated form of ERK1/2 is 4d, a cell that has organizing activity in other animals. The core of this study includes a set of chemical inhibition experiments to identify the timing of D quadrant specification as well as the signal transduction pathway and molecular signals utilized in D quadrant specification and organizing activity. Most of the interpretations are based on the drug inhibition data.*

*Perturbation of the ERK1/2 pathway uses a commercially available reagent (UO126) at multiple time intervals during early embryogenesis to perturb signal transduction and implicate this signal transduction cascade in organizing activity. Phenotypes are analyzed morphologically and by in situ hybridization using multiple molecular markers. The authors interpret the phenotype as radial and a consequence of misspecification of 4d (and of the D quadrant). This is followed by transcriptome profiling of embryos treated with UO126 or BFA and ATAC-seq to identify downstream targets of ERK1/2 signal transduction. Differential expression analysis is validated by in situ hybridization. The authors argue they have identified the receptor that specifies the D quadrant by performing chemical inhibition studies of FGFR signaling (commercially available SU5402), and demonstrating that the resulting phenotype is the same as when embryos are exposed to the ERK1/2 inhibitor. This study also includes report of a genome sequence for *O. fusiformis*, RNA-seq data for 12 developmental time points between the oocyte and the larval stage, differential gene expression analysis, ATAC-seq and developmental characterization of numerous genes at multiple developmental stages. From these data, the authors interpret *O. fusiformis* to represent the ancestral condition of spiral cleaving animals from the rationale that ERK1/2 signaling is implicated in organizer signaling in molluscs and the annelid *O. fusiformis*. The authors also state that their findings indicate that both*

autonomous and conditional mechanisms specify the D quadrant in O. fusiformis, a finding that sets it apart from other equal cleaving forms.

This study investigates an important topic through the lens of developmental biology and directly relates to the evolution of animal diversity. The spiralian represent nearly one third of animal diversity and therefore, O. fusiformis is an excellent choice of animal study system and its study is likely to reveal insights into evolution of evolution of animal body plans.

This manuscript clearly includes an impressive amount of data, some of which are more relevant to the study than others. Data for several aspects of this study are convincing. For example, chemical inhibition studies are performed with appropriate controls, range of concentrations, sample sizes, etc. The quality of in situ hybridization images is high and phenotypic interpretations are drawn from the use of numerous markers.

RESPONSE: We thank the reviewer for the positive appraisal of our manuscript.

Data included but not completely essential to this study will be useful to the community (e. g. genome sequence, RNA Seq, transcription of ERK1/2 gene, etc), but makes it challenging for the reader to follow the important thread of this study.

ACTION: Considering the reviewer's comment and to focus more the study into the essential datasets, we have removed the section about genome sequencing and annotation from the Materials and Methods and refer instead to the manuscript that characterises the genome of *O. fusiformis*, which is now available in bioRxiv (<https://doi.org/10.1101/2022.02.05.479245>). All associated datasets and that are used in this study are publicly available and properly referenced in the revised version of the manuscript.

The writing is confusing in parts, particularly the evolutionary arguments in the discussion.

ACTION: Following reviewer #1 suggestion, we have extensively modified the discussion to make its scope broader and more accessible, as well as to solidify our evolutionary claims as suggested below by this reviewer (lines 367-391).

Moreover, to accommodate all the data, the figures are complex, and some are difficult to follow or comprehend. Many figure legends do not contain sufficient detail (see detailed comments below). Together, these issues make it challenging to evaluate all of the data presented.

ACTION: We have incorporated all figure changes suggested by this and the other reviewers. Generally, we have split multi-panel figures whenever possible and sensible (which resulted in two more main figures and one more supplementary figure), re-labelled panels to make their reference in the text easier, and increased font sizes as much as possible given figure formatting requirements.

Yet, there are important pieces of data missing from this study. For example, given the large number of in situ expression patterns presented, it is surprising to not see the expression of FGFR since this is a targets of the drug SU5402. Is it in the same cell as the cell that shows activation of ERK1/2? This is an important piece of information to corroborate their SU5402 drug inhibition data.

ACTION: We have moved the expression data of *FGFR* in the coeloblastula from former Supplementary Fig. 6 to the main Figure 8 and included also other developmental stages (gastrula, elongation, and larva) in Supplementary Fig. 7d to characterise in detail the temporal and spatial expression dynamics of *FGFR* in *O. fusiformis*. The data shows that *FGFR* is expressed in the gastral plate, including 4d, at the time of specification of the D-quadrant organiser, supporting our functional observations. Later, *FGFR* is probably expressed in mesodermal and endodermal derivatives (lines 317-321).

The reviewer is not convinced that the authors can really identify 4d as the cell with organizer activity in O. fusiformis. The typical demonstration of such a function would be to perform single cell transplantation or deletion manipulations. This study does not include any such manipulations, and therefore, the authors need to update the language in the manuscript to acknowledge that their interpretations are based upon indirect deduction and that 4d 'may be' the cell with organizing activity but that future direct experimental manipulations are required to confirm.

ACTION: We agree with the reviewer that transplantation and/or deletion manipulations would be strong and direct proofs of the organising activity of the 4d cell. We have thus followed this reviewer's suggestion to tone down the text (e.g., in the abstract, line 19; and Discussion, lines 344) and written a disclaimer that further work is needed to fully confirm 4d is the organiser (lines 266-267)

The authors observe the same phenotype using all three drugs that have very different target pathways/molecules. How do the authors know that the resulting phenotype does not reflect toxicity or some other non-specific effect? Are there other chemical inhibition studies for this animal that give a distinct phenotype and could be cited? If not, demonstration of specificity is an important control to include.

ACTION: To assess for potential toxicity, we provide concentration-dependent curves for BFA and U0126 treatments and use the lowest concentrations that give the highest phenotypic penetrance (e.g., 10 μ M for U0126 and BFA). Similarly, SU5402 (FGFR inhibitor) and LY411575 (Notch pathway inhibitor) are used at the minimum concentrations that give a phenotypic outcome. We agree that BFA is a broadly unspecific drug, and thus we have toned down interpretations based on this drug, as suggested by reviewer #3 (see below). For U0126 and SU5402, we demonstrate their high specificity by assessing ERK1/2 di-phosphorylation, which is a readout of their mode of action. The fact that just a very limited number of genes are up or downregulated after U0126/BFA treatment, together with our experimental validation of a large subset of them also supports that the drug treatment is not having vast unspecific effects. In addition, we now include data on the role of the Notch pathway during *O. fusiformis* embryogenesis, which causes a different phenotype (i.e., a larva with bilateral symmetry, but an underdeveloped posterodorsal region, as expected from the expression of *delta* and *notch-like* in the D-quadrant) to those observed in BFA, U0126 and SU5402. Together, we believe these observations strongly support the specificity and validity of the functional approaches taken in our study.

Although the results presented are interesting, the proposed evolutionary argument is not strong. First, the authors argue that organizing activity in the annelid O. fusiformis is homologous to that of molluscs, but not to the annelids Platynereis dumerilli, Tubifex tubifex, Chaetopterus pergamentaceus, and Capitella teleta. Their results that implicate ERK1/2 signal transduction provides support for this argument. What is not clearly acknowledged is that the pattern of ERK1/2 activation varies across mollusc species and none of the published mollusc patterns are the same as the pattern observed in O. fusiformis. To my knowledge, the only published pattern of ERK1/2 activation that is the same as that of Owenia is found in another equal cleaving annelid, Hydroides hexagonus, an annelid within the Sedentaria clade. This calls into question the authors proposed evolutionary scenario that annelids lost ERK1/2 activation as part of D quadrant activation (Fig 6B).

ACTION: We now include a statement in the Introduction (lines 67-68) and Discussion (lines 369-372) clarifying that the identity of the cells with enriched active ERK1/2 varies among

molluscs, although in most of the studied cases occurs in the blastomeres that have organising activity (3D and 4d), as reviewed in Supplementary Table 1. Considering the reviewer's concerns, we now describe the two alternative evolutionary hypotheses (that the similarities between *O. fusiformis* and molluscs are convergent or homologous) and propose that the latter is probably more parsimonious (lines 379-388).

Second, a stronger evolutionary argument would include multiple pieces of evidence. For example, the cellular mechanism of D quadrant specification is known for some molluscs (e. g. Patella and Lymnea) and involves critical cell contacts between the macromeres and first quartet micromeres. At a minimum, this literature should be discussed.

ACTION: We now include this matter in the discussion (lines 375-379).

There are numerous additional issues that need to be addressed and are outlined below.

Detailed comments:

-The authors should acknowledge that unequal cleavage may be ancestral for annelids. Even if the authors disagree, the published alternate opinion should be cited (Dohle 1999).

ACTION: As suggested, we now include a mention and reference to Dohle's chapter (line 53).

-Line 29, the wording 'instructing role of ERK1/2...' is misleading. Activation of ERK1/2 occurs when a cell receives a signal and is NOT a direct sign of a signal being given or 'instructing'

-Line 38, 'instruct neighbouring tissues' should be 'instruct neighbouring cells'. Tissues have not yet formed at this stage.

-Line 83 - 85, overstatement to say that ERK1/2 activity spread convergently to some annelid lineages. Best existing candidate of a convergent mechanism is Hydroides and to my knowledge, there is no functional data for Hydroides (if there is, please cite and add statement) a requirement to make a statement about function (e.g. 'activity').

RESPONSE: All the above points refer to text in an older version of the introduction (as in the manuscript version uploaded to bioRxiv) and absent from the submitted and revised version.

-Line 92, citation 32, is this the correct citation since the title suggests that the study is about two other annelid species. Is it really about O. fusiformis?

RESPONSE: Yes, the study includes data on *O. fusiformis* as well as two other annelid species.

-Line 101, why show transcript expression for ERK1/2 ortholog when the relevant information is whether or not ERK1/2 is phosphorylated?

ACTION: We agree with the reviewer that phosphorylation of ERK1/2 is more functionally relevant than the transcriptional dynamics of the gene. However, we also find an upregulation of ERK1/2 expression from 4 hpf onwards, which is relevant because it is consistent with the enrichment dynamics of di-P-ERK1/2 observed by immunohistochemistry. We now make this point clearer in the first section of the Results (lines 113-115).

-The authors identify a single cell that is di-P-ERK1/2 positive (Fig 1D, Fig 2B). How do the authors really know it is 4d if it is an equal cleaving form? The size of 4qs appear to be the same (Fig 1E).

ACTION: We have entirely rewritten the first section of results and simplified former Figure 2 into a main Figure and a Supplementary Figure to better explain the reasoning and experimental approach to identify the D-quadrant and 4d blastomere, as suggested by reviewer #1. Briefly, we infer the 4d cell identity based on the delayed cleavage of one of the 4q micromeres with respect to the others of the same tier and its subsequent division into two larger cells that remain undivided during gastrulation (which is like the dynamics of 4d described in other annelids), as

well as the enrichment in active ERK1/2 in that 4q micromere, as also observed in *Hydroides* and other molluscs.

In addition, the larger size of the z-stack projection of the vegetal pole of a 6 hpf embryo in Supplementary Fig. 1f better illustrates the observation that 4d is larger than any of the daughter cells derived from 4a–c micromeres at that time point.

-Line 121, are the two daughter cells of 4d di-P-ERK1/2 positive? Or is signal lost once 4d divides? How long does the di-P-ERK1/2 form persist in 4d or its daughters?

ACTION: We now include vegetal views of di-P-ERK1/2 immunohistochemistry from 7 to 9 hpf in Figure 6a. At those stages, active ERK1/2 persists in MR and ML (4d daughter cells) at least until gastrulation is completed (9 hpf). MR and ML cleave after 9 hpf and ERK1/2 activation becomes more difficult to track (line 229-231).

-Lines 124 – 126, Cannot tell from wording what exactly ‘cellular dynamics’ refers to.

ACTION: We have removed this sentence when rewriting the first section of the Results.

-Lines 149 - 152, exposure to UO126 is from 0.5 – 5hr, which encompasses 3 phases of di-P-ERK1/2 positive cells: animal pole cells, then ‘4d’ then 4d plus other vegetal 2q cells. Therefore, wording of this statement is an oversimplification since the resulting phenotype is likely a reflection of interfering with signaling at all three phases. Please expand statement to acknowledge that e. g. the apical tuft phenotype may be a consequence of inhibiting activation of P-ERK1/2 in the 1q111 cells and other defects may reflect inhibition of 2q12 and 2q22 cells, perhaps some of the posterior structures.

ACTION: We have amended the text as follows: “Therefore, activation of ERK1/2 signalling in the 4d cell at the coeloblastula stage is required to specify and develop posterior and dorsal structures during *O. fusiformis* embryogenesis, although inhibition of ERK1/2 activity in the 1q111 (at 4 hpf) might also contribute to the reduced apical organ phenotype.” (lines 143-147). We do not mention the 2q12 and 2q22 cells because ERK1/2 activity in these cells occurs at 6 hpf, after the window of treatment (0 to 5 hpf).

-Line 160, what does a ‘compressed morphology’ mean? Compressed along what axis? This is not a standard term and therefore it needs to be defined.

ACTION: We have changed the text to clarify that “compressed” refers to the apical-ventral axis (lines 155-156). In addition, and as suggested by reviewer #1, we now define each phenotype (including the compressed one) based on a set of morphological and molecular landmarks, which we specify in the Materials and Methods (lines 528-545) and in Supplementary Table 3. We also include diagrams of these landmarks next to the main phenotypic characterisation in Supplementary Fig. 3d, as suggested by reviewer #1.

-Line 169, ‘ERK1/2 activity is required for normal embryonic activity’... this does not fit very well with the pattern of activation as shown by immunohistochemistry. How do the authors explain this?

ACTION: This sentence now reads as: “Therefore, ERK1/2 activity is essential for normal embryonic patterning and posterodorsal development throughout most spiral cleavage in *O. fusiformis*.” (lines 163-165).

-Line 202, the following developmental genes are not to my knowledge involved in axial patterning in spiralian: six3/6, gsc, cdx, AP2, foxQ2. Need to add rationale, supporting evidence or modify language to state that these genes are markers of particular cell and tissue types.

ACTION: We have re-worded this section as suggested by the reviewer: “including a variety of transcription factors that are markers of particular cells and tissue types (e.g., *six3/6*, *gsc*, *cdx*, *AP2*, *foxQ2*), required for mesoderm development...” (lines 195-197).

-Line 202, ‘recurrently’ is not a word.

ACTION: We have removed the word.

-Line 241, this statement does not obviously follow from the previous information in the paragraph ‘ERK1/2 di-phosphorylation in 4d seems to delay cell cycle progression in this cell with respect to 4a–c’. How does ATAC sequence data relates to cell cycle progression?

ACTION: We removed the reference to cell cycle progression: “Therefore, ERK1/2 di-phosphorylation in 4d seems to control the activity of transcriptional regulators that induce posterior fates (*cdx*) and cell-cell communication genes (*delta*).” (lines 239-241).

-Line 249 – 250, how is the fate of 3q cells in an echiuran relevant to this study?

ACTION: We now clarify this point: “The transcription factor *foxH* (Supplementary Fig. 6e), which regulates mesoderm development during gastrulation in vertebrate embryos³⁸⁻⁴¹, is detected in four micromeres adjacent to 4d (Fig. 5b, c), which might contribute to lateral ectomesoderm according to lineage tracing studies in the annelid *Urechis caupo*³⁵.” (lines 244-248).

-Line 262, 4d does not appear to be larger than the other 4q cells (Fig 1E) as stated in the text. A clear image showing this needs to be provided to make such a claim.

ACTION: As indicated above, we moved former Figure 2e to a stand-alone Supplementary Figure (1), which allowed us to make panels and labels larger. The larger size of the undivided 4d cell compared to the daughter cells of 4a–c at 6 hpf is now more evident (Supplementary Figure 1f).

-Line 339, add ‘6hr’ to figure reference ‘Figure 1E’. It is challenging to find information the figures referenced in the text since unique identifiers of individual panels are lacking. This is a general challenge throughout the figures.

ACTION: We have changed the reference accordingly. As indicate above, we have tried to break multi-panel figures into individual figures and reference each panel individually whenever possible to improve readability.

-Line 354, ‘...indicating that unlike most annelids’ is an overstatement. The manuscript cited is that of a single species. Although there may be data for 1 – 2 other species, it is prudent for the authors to acknowledge that there are almost 20,000 described annelids. Change ‘most’ to ‘some’.

ACTION: Changed as suggested.

-Line 486, Add a reference to support the agar plug DNA extraction method.

ACTION: As indicated above, we have removed mention to accessory datasets in Materials and Methods (e.g., genome sequencing and annotation), which are now accessible in DOI <https://doi.org/10.1101/2022.02.05.479245>.

-Line 980, ‘..BFA and UO126 treated larva’ Should be ‘larvae’. This is incorrect as worded since the larvae themselves were not exposed to the drug. The reviewer’s understanding is that embryos were treated with drug and then raised to larval stages for phenotypic analysis.

ACTION: We have amended the text in Supplementary File to “Volcano plots for BFA and UO126 treated embryos, studied at 24hpf (larval stage).”.

Figures:

There is a lack of a unique identifier (e. g. panel letter or label) for each image/graph/schematic presented. As a reviewer, it is difficult to refer to individual parts of figures since one letter may refer to up to 14 different images.

ACTION: See points above (we have included unique identifiers to panels whenever possible).

Figure 1B, recommend including the annelids specifically mentioned in the text in the phylogenetic tree to give the reader a better appreciation of their phylogenetic distribution and interpretations that are drawn. For example, representatives of both Sedentaria and Errantia are cited, as well as the early branching species Chaetopterus (reviewed in Weigert and Bleidorn 2016).

ACTION: Figure 1c includes these taxa.

Figure 2, what are the inserts in panels shown in B? in D? please add info to legend. Same question for Figure S2C. Check all figure legends.

ACTION: We have amended all figure legends to make them more informative and explanatory, addressing these and all other points raised.

Fig 2C, 2D, recommend adding ventral view insets for the control images to compare with the ventral views of the treated animals.

Fig 2D Chaetae should be labeled since they are mentioned as one of the scored characters in the text.

ACTION: We changed these two points as suggested.

Fig. 2E, right, observed larval phenotype graphic is very difficult to understand! There is almost no text in the legend that explain this graphic and how it relates to the left part of the panel.

ACTION: We have amended the legend to clarify this panel, while maintaining the caption length below 350 words as per the journal style guidelines.

Fig. 2E, far fewer cases (n=) for critical time interval (4 - 6 hr) than for other time intervals. Why? Also, do these few cases represent independent replicates? Please add to legend that number of cases is shown to the right of each bar specifying the time interval.

ACTION: We used available samples from independent replicates conducted last summer to increase the number of cases for the critical time interval (4 – 6 hpf) and merged all phenotypic assessments done for the window 0 – 5 hpf, to better reflect the large number of embryos that were analysed. Detailed numbers are also provided in Supplementary Table 5. In addition, we have added the sentence “Number of cases is shown to the right of each bar specifying the time interval.” to all instances that apply (legends for Figure 3, 7 and 8).

Fig 3B, Y axis label is confusing ‘no differentially expressed genes’ should be ‘# differentially expressed genes’ or ‘No. differentially expressed genes’

ACTION: We amended this issue as suggested by reviewer #1.

Fig 3B, what do the circles, grey circles, and lines grey connecting lines at the bottom of the figure panel denote? Add heading and/or info to legend. What are the exact pairwise comparisons that result in up and down regulated genes shown for each of the bars in the bar graph?

ACTION: We amended the legend accordingly (lines 838-839). The exact comparisons are indicated by the grey solid dots in the bottom part of the plot.

Fig 3C, how do the authors explain the substantial differences in identity of differentially expressed genes when embryos are treated with BFA versus UO126 with the same resulting phenotype? This needs to be addressed.

ACTION: As highlighted by reviewer #3 (see below), BFA generally impairs protein trafficking and thus has more off-target effects (hence the substantial differences in differentially expressed genes). The phenotypic similarities between BFA and U0126 are probably explained by the fact that both direct inhibition of ERK1/2 di-phosphorylation and of a likely upstream protein secretion event results in 4d miss-specification and failure to produce a bilaterally symmetrical embryo, which causes evident and defined morphological changes in *O. fusiformis*. However, we do not exclude that there might be differences between BFA and U0126 treated embryos at other levels and with respect to other developmental processes that would fall beyond the remit of our study. We now mention this issue (lines 187-190) and we have also toned down the interpretation of BFA results throughout the manuscript (as suggested by reviewer #3).

Fig 3D, what is the coding of the colored dots on the left side of the panel? What does inclusion of a dot indicate?

ACTION: Coloured dots indicate the treatment condition at which each of the genes is differentially expressed (each condition has a different colour). We amended the legend to clarify this point (lines 843-846).

Fig 3D, Should be 'vertical' dotted line, not horizontal dotted line as indicated in legend. Perhaps this refers to a previous version?

ACTION: Amended accordingly.

Fig 4, hard to see cell boundaries (AcTub) in fluorescent panels in A, B and C. Better in some panels in F.

ACTION: We have amended these images to increase the signal at the cell boundaries (for the *cdx* image in Figure 5b, that we considered the poorest).

Fig 4D, are cdx and delta the only developmental genes identified with open chromatin in this ATAC-seq analysis? If so, please state in text. If not, why were these genes highlighted and not others?

ACTION: Open chromatin regions are abundant in embryos at 5 hpf, as now described in DOI <https://doi.org/10.1101/2022.02.05.479245>. We focused on *cdx* and *delta* as these are the two candidate genes uniquely expressed in 4d cell and could thus inform of the regulatory cascade linking ERK1/2 di-phosphorylation and the activation of downstream targets. We have clarified the text about ATAC-seq data according to reviewer #3 comments (see below).

Fig 4E is difficult to understand and there is not sufficient explanation in the legend. Why is the heading for the schematic 'no cellular contacts'? It appears that the point of the schematic is cellular contacts?

ACTION: We have amended the axis label to read as “No. of cellular contacts” and changed the plotting style to bar plots to improve data visualisation.

Fig 5C, right side of panel is very hard to decipher. Needs explanatory text. Same issue as in Fig. 2E

ACTION: We have amended the figure legends of both figures to clarify what the right side of both panels depict (lines 820-821, lines 910-911 and lines 941-942).

Fig 6 legend Lines 936 – 7, activation in 4d by ERK1/2 limits an anteriorizing signal in the A – C quadrants. Rather than signaling from 4d, is it possible that the expansion of gsc expression is a more direct affect of the interference of di-P-ERK1/2 in the 1q111 cells?

ACTION: Thanks for raising this point; we did not think of this possibility. We now mention in the Discussion that the signal controlling *gsc* expression (and hence its expansion when blocking the specification of the 4d cell) might come from the apical organ (lines 354-357). Whether this

is a direct effect of interfering di-P-ERK1/2 activity in those cells is unclear, since *gsc* is not significantly differentially expressed in our datasets. As we point out, further work is needed to dissect the exact molecular mechanisms responsible for the earliest signalling events driving axial polarity in *O. fusiformis* and spiralian generally (lines 388-391).

Fig S1C, is this really low level activation in early cleavage stages or lack of specific signal? What is needed are negative controls such as a no primary antibody and no antibody to see signs of any endogenous reaction or background.

ACTION: We thank the reviewer for the suggestion. We now include images of negative control conditions (no primary antibody) in Supplementary Fig. 2d. Negative control is similar to the condition that includes the primary antibody and thus we cannot discard that the observed signal at early stages is unspecific. We have amended the text in the second paragraph of the Results accordingly (lines 110-124).

-Fig S2C, BFA treatment, looks like see expansion of neural tissue? How do the authors interpret this phenotype? Is this related to the di-P-ERK1/2 in the 1q111 cells?

ACTION: In BFA-treated radialised embryos, *six3/6* expression is only in the endoderm (foregut/midgut connection), while the wild-type *six3/6* apical ectodermal domain is absent/reduced. We now clarify this point in the legend (Supplementary Information).

Fig S4, what is the rationale for organizing the gene expression patterning in the groupings shown? Need to add reference to individual panel letters to provide this information.

ACTION: Gene expression is organized based on the stage(s) at which genes are differentially expressed between control and treated conditions. We have amended the figure legend to provide this information and included unique identifiers to each condition (Supplementary Information).

Fig S5B, What views of the embryos are shown in the panels?

ACTION: Main panels show ventral views and insets are lateral views. We amended the text accordingly (figure legend to Supplementary Figure 6, Supplementary Information).

Fig S5B, in legend it is stated that foxH is expressed in cells giving rise to mesodermal derivatives. Does this mean that these are daughters of 4d, possibly ML and MR, which generate most mesoderm in spiralian. If so, is this not a contradiction with a statement that 4d does not divide during this time interval?

ACTION: Thanks for raising this point. We now include in Figure 6 (panels d and e) a detailed analysis of *foxH* expression during gastrulation (from 7 to 9 hours post fertilisation). This new data demonstrates that this gene is not expressed in MR and ML, which do not cleave during these time points (as shown in Supplementary Fig. 1e, f and Figure 6a). We hypothesise that *foxH* positive cells form lateral ectomesoderm (consistent with the potential fate of the *foxH* expressing blastomeres in *Urechis*, see point above; lines 248-250 in the main text). Further analyses tracing the division dynamics of MR/ML after 9 hpf are required to decipher how these two cells contribute to the trunk endomesoderm.

Fig S6C, should be 'vertical' dotted line, not 'horizontal' dotted line.

ACTION: Amended as suggested.

Fig S6F, legend says that there is reduction of six3/6 expression but the images actually show an expansion in the area of expression. Please correct/explain.

ACTION: We now clarify this point (Supplementary Fig 7, Supplementary Information, figure legend: "Radial larvae are phenocopies of BFA and U0126 radial larvae with reduction of apical markers (*six3/6*) in the apical ectoderm..."). The expression of *six3/6* in the apical ectoderm (i.e.,

apical organ) is absent in treated embryos versus control, while *six3/6* expression in the foregut/midgut connection is present in both conditions.

Reviewer #3 (Remarks to the Author):

*This is a sound study that will be of interest to specialists within the field of evolutionary developmental (evo-devo) biology. The work provides evidence that FGFR and ERK1/2 signaling specify and control the activity of the embryonic organizer in the spiral-cleaving annelid *O. fusiformis*, which undergoes a conditional mode of embryonic development. The paper is beautifully written. The authors conduct precise gene expression analyses at early stages of development to identify where and when phosphorylated Erk is present, followed by pharmacological inhibition of Erk activity to assess gene expression and morphological effects. The RNA-Seq data are independently validated with RNA ISH. Lastly, pharmacological inhibition of FGFR phenocopies the effects observed upon drug inhibition of Erk. Enough detail is provided in methods for the work to be reproduced by others. Most of the data support the conclusions of this study.*

RESPONSE: Thank you for the positive appraisal of our work.

However, additional evidence is needed to strengthen some of the key claims, as outlined below:

-Figure 2 describes a detailed expression analysis for di-phosphorylated ERK1/2. Because only cell positional criteria were used to infer the identity of the 4d cell, additional molecular markers for 4d could be used to strengthen the point that di-phospho ERK1/2 is enriched in 4d.

ACTION: Thanks for raising this important point. Following reviewer #1 suggestion, we have rewritten the first section of the Results to clarify that the identity of the 4d blastomere is inferred based on both cell positional criteria and ERK1/2 activity. Unfortunately, and as we now show in Supplementary Table 9, there are not many known molecular markers of 4d in spiralian (because the gene regulatory network downstream of ERK1/2 activation was undescribed until our study). Importantly, *cdx* and *delta* are amongst the handful of genes reported to be expressed in the 4d in other molluscan and annelid embryos, as they are in *O. fusiformis* too, further supporting our initial cell identity inference for the 4d blastomere. We now clarify this point in the main text (lines 225-227).

*-The analysis of the two drugs, BFA and U0126, is detailed and well-presented. However, drugs can have numerous off-target effects. Is there a way in *O. fusiformis* to disturb the phosphorylation of ERK1/2 by either using a masking antibody against the phosphorylation site (s), or RNAi (shRNA) to downregulate MEK?*

RESPONSE: We thank the reviewer for the suggestion. Gene-specific functional approaches (e.g., shRNAs) are not yet established in *O. fusiformis* and thus drugs are the only functional approach currently available in this non-model annelid species. While we acknowledge the potential off-target effects of the drugs we use (and hence we have toned down the text accordingly, in particular for BFA, see point below), pharmacological inhibition also allows us to assess different time windows of functional perturbation, which is essential to identify the exact moments of action (and the different roles throughout development) of the FGFR/ERK/Notch signalling pathways.

-Lines 216-222: The conclusions drawn are incorrect for two reasons. First, ATAC-Seq provides an assessment of open chromatin at specific regions. This method only allows inferences to be made about which transcription factor motifs are present in a specific region of DNA that is

“open”. ATAC-Seq cannot identify transcription factors bound to the accessible regions. ChIP-Seq can do this.

ACTION: We have amended the text (lines 235-239): “To investigate how ERK1/2 might control activation of *cdx* and *delta*, we used Assay for Transposase-Accessible Chromatin using sequencing (ATAC-seq) data at 5 hpf to identify transcription factor motifs present in accessible chromatin regions associated with these two genes (Fig. 5d). These include motifs of transcriptional regulators known to be modulated by ERK1/2 phosphorylation, such as ETS, RUNX and GATA factors.”

*Second, to test causality and the author’s claim in line 221 (ERK phosphorylation in 4D seems to control the activity of transcriptional regulators) the ATAC-Seq experiments needs to be conducted in the presence of U0126, or other more specific phospho-ERK manipulations proposed in point #2. According to the authors hypothesis, an expected outcome would be that the open chromatin regions of *cdx* and *delta* will be “closed” upon U0126 treatment.*

RESPONSE: We thank the reviewer for the suggestion and agree that a comparative study of ATAC-seq data between control and treated conditions would provide clearer insights on the exact regulative mechanisms that induce activation of downstream targets in 4d after ERK1/2 di-phosphorylation. However, following the editor’s indication and the fact that we do not have access to live embryos of *O. fusiformis* during winter months, we kept these experiments for future studies.

*-A central pathway in this paper appears to be the Notch pathway, as *delta* is downstream of FGFR/ERK. Pharmacological inhibitors for Notch are available and readily used in evo-devo projects. They authors are encouraged to used them and test whether Notch inhibition partially phenocopies the effects observed upon FGFR/ERK inhibition.*

ACTION: Following the reviewer’s suggestion, we now include a detailed characterisation of the pharmacological inhibition of the Notch-Delta pathway in *O. fusiformis* before and after the specification of 4d using the gamma-secretase inhibitor LY411575 (Figure 7). These sets of experiments were conducted during the summer months of 2021 (breeding season of *O. fusiformis*) as follow-up efforts of the lab to study the role of Notch during the development of *O. fusiformis*. Because *delta* and a *Notch-like* receptor are downstream of ERK1/2 and expressed in 4d and posterodorsal ectoderm, our functional experiments test the hypothesis that inhibition of Notch-Delta pathway should affect neither the establishment of the bilateral symmetry nor the formation of hindgut tissues (because Notch-Delta is downstream of the specification of the D-quadrant organiser) but might affect the development of the posterodorsal region and impair axial growth (because of the expression of Notch-like receptor and the evolutionary conserved role of this pathway in posterior elongation, as reviewed in Supplementary Table 14). As shown in Figure 7, treatment with LY411575 supports this hypothesis, with embryos treated from 0 to 5 hours post fertilisation showing normal morphology (hence Notch-Delta is not upstream of ERK1/2), but treatment from 5 hours to larvae causing defects in posterodorsal development without compromising the establishment of the bilateral symmetry and *cdx* (hindgut marker) expression. Altogether, this new dataset (described in lines 285-307) supports the originally proposed model, although further analyses will be required to explore in greater detail the exact mode of action of the Notch-Delta pathway during the development of *Owenia* and in relation to the potential organising/inductive role of ERK1/2 in the 4d cell and neighbouring cells.

*-The claims related to BFA treatment should be toned done throughout the paper, as BFA is a global protein trafficking inhibitor that affects many many proteins. For example, in line 251-252 it is stated that *gsc* expression requires inductive signals based on the BFA data. However,*

the trafficking of proteins made from maternally deposited mRNAs will also be affected by BFA, hence the requirement of autonomous signals cannot be ruled out for gsc expression.

ACTION: Thanks for highlighting this important point. In this revised version of the manuscript, we have generally toned down the conclusions reached through the study of BFA treatment. We also indicate the possibility that cell-autonomous mechanisms drive gsc expression (lines 356-357).

-The data with the FGF inhibitor make the compelling argument that FGFR is upstream of phosphor-ERK. However, these data could be presented in more detail. For example, it would help the reader if the phenocopy of the SU5042 and U0126 treatments is discussed in more detail - morphological and gene expression effects can be compared and contrasted. Again, the BFA treatment data should be toned down here due to its broad effects on protein trafficking.

ACTION: As suggested by the reviewer, we have expanded the comparisons between the phenotypes observed after U0126, SU5042 and LY411575 (lines 297-301 and 324-334). Moreover, following the points raised by reviewer #1, we now include schematic drawings of the phenotypes highlighting the morphological and molecular landmarks that we used to characterise all phenotypes, as well as a Supplementary Table 3 with the scores for these criteria and a brief description of the approach taken to described and compare phenotypes in Materials and Methods (lines 528-545).

Reviewers' Comments:

Reviewer #1:

Remarks to the Author:

The authors have properly address all the reviewer concerns. The manuscript is now more focused, with improved flow and Figures. I am happy to see all the effort by the authors in improving the text as well.

Reviewer #2:

None

Reviewer #3:

Remarks to the Author:

The authors have addressed all my concerns. The revised manuscript now better communicates to a broader audience the key conclusions of this study. The revised text, figures and legends are very much improved, making this work an impactful contribution to the field.

Response to reviewers

Reviewer #1 (Remarks to the Author):

The authors have properly addressed all the reviewer concerns. The manuscript is now more focused, with improved flow and Figures. I am happy to see all the effort by the authors in improving the text as well.

RESPONSE: Thanks.

Reviewer #3 (Remarks to the Author):

The authors have addressed all my concerns. The revised manuscript now better communicates to a broader audience the key conclusions of this study. The revised text, figures and legends are very much improved, making this work an impactful contribution to the field.

RESPONSE: Thanks.